# ON LEARNING READ-ONCE DNFS WITH NEURAL NETWORKS

## ABSTRACT

Learning functions over Boolean variables is a fundamental problem in machine learning. But not much is known about learning such functions using neural networks. Because learning these functions in the distribution free setting is NP-Hard, they are unlikely to be efficiently learnable by networks in this case. However, assuming the inputs are sampled from the uniform distribution, an important subset of functions that are known to be efficiently learnable is read-once DNFs. Here we focus on this setting where the functions are learned by a convex neural network and gradient descent. We first observe empirically that the learned neurons are aligned with the terms of the DNF, despite the fact that there are many zero-error networks that do not have this property. Thus, the learning process has a clear inductive bias towards such logical formulas. To gain a better theoretical understanding of this phenomenon we focus on minimizing the population risk. We show that this risk can be minimized by multiple networks: from ones that memorize data to ones that compactly represent the DNF. We then set out to understand why gradient descent "chooses" the compact representation. We use a computer assisted proof to prove the inductive bias for relatively small DNFs, and use it to design a process for reconstructing the DNF from the learned network. We proceed to provide theoretical insights on the learning process and the optimization to better understand the resulting inductive bias. For example, we show that the network that minimizes the $l_2$ norm of the weights subject to margin constraints is also aligned with the DNF terms. Finally, we empirically show that our results are validated in the empirical case for high dimensional DNFs, more general network architectures and tabular datasets.

## 1 INTRODUCTION

The training objective of overparameterized neural networks is non-convex and contains multiple global minima with different generalization properties. Therefore, just minimizing the training objective does not guarantee good generalization performance. Nonetheless, neural networks trained in practice with gradient-based methods show good test performance across numerous tasks (Krizhevsky et al., 2012; Silver et al., 2016), suggesting an *inductive bias* towards desirable solutions. Understanding this inductive bias and how it depends on the algorithm, architecture and data is one of the major open problems in machine learning (Zhang et al., 2017; Neyshabur et al., 2018).

In recent years, there have been major efforts to tackle this challenge. One line of works considers the Neural Tangent Kernel (NTK) approximation of neural networks which reduces to a convex optimization problem (Jacot et al., 2018). However, it has been shown that the NTK approximation is limited and does not accurately model neural networks as they are used in practice (Yehudai & Shamir, 2019; Daniely & Malach, 2020). Other works tackle the non-convexity directly for specific cases. However, current results are either for very simplified settings (e.g., diagonal linear networks Woodworth et al., 2019) or for specific cases such as regression with 2-layer models and Gaussian distributions (Li et al., 2020), or for impractical settings with infinitely wide two-layer networks (Chizat & Bach, 2020). Presumably, the reason for this relatively limited progress is the lack of general mathematical tools to analyze the non-convexity directly, except for a few simplified cases.

One approach to make progress on this front is to use empirical tools in addition to theory, when the theoretical analysis is not tractable. In this work, we use this approach and study the inductive bias

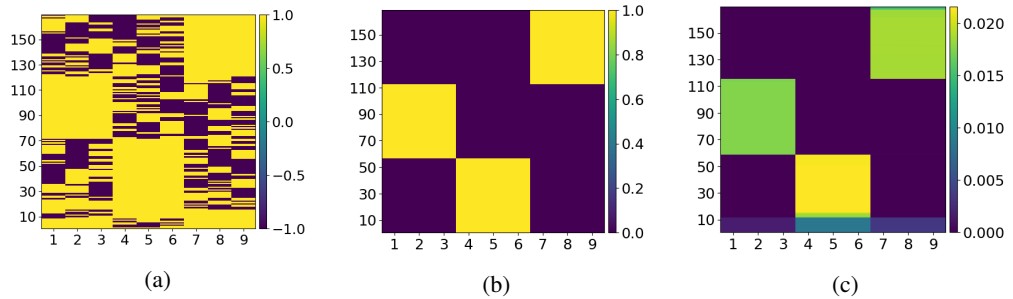

Figure 1: Examples of global minima for learning the read-once DNF: $(x_1 \wedge x_2 \wedge x_3) \vee (x_4 \wedge x_5 \wedge x_6) \vee (x_7 \wedge x_8 \wedge x_9)$ with a convex network. (a) Global minimum that memorizes the training points. (b) Global minimum that recovers the DNF. (c) Global minimum that GD converges to.

in a challenging and novel setting which is not addressed in previous theoretical works. Concretely, we consider learning read-once DNFs under the uniform distribution with a one-hidden layer, non-homogeneous convex network with ReLU activations and gradient descent (GD).[1]

In computational learning theory, the problem of learning DNFs has a long history. Learning DNFs is hard (Pitt & Valiant, 1988) and the best known algorithms for learning DNFs under the uniform distribution run in quasi-polynomial time (Verbeurgt, 1990). On the other hand, for learning *read-once* DNFs under the uniform distribution there exist efficient learning algorithms (Mansour & Schain, 2001).[2] Therefore, it is interesting to understand whether neural networks can learn read-once DNFs under the uniform distribution and this motivates the study of the inductive bias in this case.

To better understand the inductive bias, we focus on the population setting. We show that even in this setting, where the training set consists of all possible binary vectors, there exist global minima of the training objective with significantly different properties. For example, a global minimum which memorizes the training points in its neurons, and another minimum whose neurons align exactly with the terms of the DNF, which we call a DNF-recovery solution. Figure 1a-b shows an example of these global minima. Therefore, the key question is what is the inductive bias of GD in this case. Namely, to which global minimum does it converge?

To address this question, we provide a computer-assisted proof for the convergence of GD in low dimensional DNFs. We circumvent the difficulty of floating point errors in the computer-assisted proof by utilizing a unique feature of our setting that allows to perform calculations in integers. We prove that under a symmetric initialization, the global minimum that GD converges to is similar to a DNF-recovery solution. Figure 1c shows an example of the global minimum GD converges to, which indeed looks similar to the DNF-recovery solution in Figure 1b. We prove, using the computer assisted proof, that after a simple procedure of pruning and rounding of the weights, we can obtain the exact DNF-recovery solution from the model that GD converges to. Consequently, the terms of the DNF can be reconstructed from the network weights.

We provide additional theoretical results for the population setting. We show that for a symmetric initialization, gradient descent has the following unique stability property: if at some iteration a neuron is aligned with a term of a DNF, it will continue to be aligned with the term for all subsequent iterations. This gives further evidence that GD is biased towards neurons that are aligned with terms. We also study minimum $l_2$ norm solutions of our problem, inspired by recent works that show connections between norm minimization and GD for homogeneous models (Lyu & Li, 2020; Chizat & Bach, 2020). We prove that the $l_2$ minimum norm solutions are all DNF-recovery solutions.

We corroborate our findings with empirical results which show that our conclusions hold more broadly. Specifically, we perform experiments on DNFs of higher dimension, standard one-hidden layer neural networks and Gaussian initialization. Taken together, our results demonstrate that gradi-

---

[1] Non- homogeneity here is a result of a bias in the second layer.

[2] In a read-once DNF each literal appears at most once. See Section 3 for a formal definition.

ent descent can recover simple descriptions of Boolean functions from data, a fact that has important implications on the question of interpretability.

## 2 RELATED WORK

Recently, several works studied the inductive bias of two-layer homogeneous networks and show connections between gradient methods and margin maximization (Chizat & Bach, 2020; Lyu & Li, 2020; Ji & Telgarsky, 2020). Preliminary results for non-homogeneous networks were provided in Nacson et al. (2019). We note that these results do not provide convergence guarantees and hold under assumptions on the dynamics of gradient methods (e.g., that they reach a certain loss value). Other works study fully connected neural networks under certain assumptions on the data such as linearly separable data (Brutzkus et al., 2018) or Gaussian data (Safran & Shamir, 2018). Malach & Shalev-Shwartz (2019) show that certain structured Boolean circuits can be learned with a network architecture that is specialized for their data structure. Fully connected networks were also analyzed via the NTK approximation (Du et al., 2019; 2018; Arora et al., 2019; Fiat et al., 2019).

Another line of works (Saad & Solla, 1996; Goldt et al., 2019; Tian, 2019) studies neural networks in a student-teacher setting and shows a "specialization" phenomenon, where a subset of student neurons aligns with teacher neurons. The main difference from our setting is that we consider classification on binary data, and they consider regression tasks on non-discrete data (e.g., Gaussian). Furthermore, we perform exact DNF recovery which is unique in our classification setting.

Bengio et al. (2006); Bach (2017) study convex networks with infinitely many hidden units and devise convex relaxation algorithms. In this work, we consider convex networks with finitely many hidden neurons and study gradient descent on a non-convex objective.

Rudin (2019) argues that methods for explaining large neural networks should be avoided because networks are too complex for humans to understand. However, we show, albeit in a restricted setting, that learned networks can be rather simple, and are easily mapped to the underlying DNF.

## 3 PROBLEM FORMULATION

**DNFs and Read-Once DNFs:** In what follows, we use $[n]$ to denote the set $\{1, 2, ..., n\}$, and let $\mathcal{X} = \{\pm 1\}^D$ and $\mathcal{Y} = \{\pm 1\}$. DNFs (e.g., see O'Donnell, 2014) are usually defined on inputs with entries in $\{0, 1\}$ to an output in $\{0, 1\}$. In this work, we consider DNFs on inputs with entries in $\{\pm 1\}$ and output in $\{\pm 1\}$. Therefore, we will use the following notation for DNFs. Let $\boldsymbol{t}_1^*, ..., \boldsymbol{t}_K^* \in \{0, 1\}^D$. We refer to each $\boldsymbol{t}_n^*$ as a *term* and define its set of active indices by $\mathbb{A}_n = \{j \in [D] \mid t_{nj}^* = 1\}$, where $t_{nj}^*$ is the $j$th entry of $\boldsymbol{t}_n^*$. We define a DNF $f^* : \mathcal{X} \to \mathcal{Y}$ with $K$ terms $\boldsymbol{t}_1^*, ..., \boldsymbol{t}_K^*$ as follows: $f^*(\boldsymbol{x}) = 1$ if $\exists n \in [K]$ $s.t.$ $\boldsymbol{x} \cdot \boldsymbol{t}_n^* = |\mathbb{A}_n|$, and otherwise $f^*(\boldsymbol{x}) = -1$. Notice that $f^*$ is monotone.

We refer to $\boldsymbol{t}_n^*$ as term $n$ of $f^*$ and we say that a sample $\boldsymbol{x} \in \mathcal{X}$ satisfies the term $\boldsymbol{t}_n^*$ if $\boldsymbol{x} \cdot \boldsymbol{t}_n^* = |\mathbb{A}_n|$. We refer to $|\mathbb{A}_n|$ as the size of the term $\boldsymbol{t}_n^*$. To compare our notation with the standard one, for example, the DNF $(x_1 \wedge x_2) \vee (x_3 \wedge x_4)$ with 4 inputs has terms $\boldsymbol{t}_1^* = (1, 1, 0, 0)$ and $\boldsymbol{t}_2^* = (0, 0, 1, 1)$. We will use the standard notation when convenient, e.g., as in Figure 1. In this work we will focus on ***read-once*** DNFs where for all $i \neq j \in [K]$, $\mathbb{A}_i \cap \mathbb{A}_j = \varnothing$

**Learning Setup:** Let $\mathcal{D}$ be a the uniform distribution over $\mathcal{X}$ and $f^*$ be a monotone *read-once* DNF......[3]. We consider learning $f^*$ given a training set $\mathbb{S} \subseteq \mathcal{X} \times \mathcal{Y}$, where for each $(\boldsymbol{x}, y) \in \mathbb{S}$, $\boldsymbol{x}$ is sampled IID from $\mathcal{D}$ and $y = f^*(\boldsymbol{x})$. Denote $\mathbb{S}_x = \{\boldsymbol{x} \mid (\boldsymbol{x}, y) \in \mathbb{S}\}$ and the positive samples by $\mathbb{S}_p = \{\boldsymbol{x} \mid (\boldsymbol{x}, 1) \in \mathbb{S}\}$.

**Neural Architecture:** We consider a **convex** one-hidden layer neural network (NN) with $r$ hidden units and parameters $(\boldsymbol{W}, \boldsymbol{b}) \in \mathbb{R}^{rD} \times \mathbb{R}^r$ which is defined by:

$$N(\boldsymbol{x}; \boldsymbol{W}, \boldsymbol{b}) = \sum_{i \in [r]} \sigma(\boldsymbol{w}_i \cdot \boldsymbol{x} + b_i) - 1 \tag{1}$$

---

[3]In the case of the uniform distribution, we can assume monotone DNFs WLOG. This follows since for the uniform distribution (that has IID $Bernoulli(0.5)$ entries), by symmetry, any negated literal can be replaced with the original literal (without negation) and all our results still hold.

where $\sigma(x) = max\{0, x\}$ is the ReLU function, $\boldsymbol{w}_i$ is the $i^{\text{th}}$ row of $\boldsymbol{W}$ and $b_i$ is the $i^{\text{th}}$ entry of $\boldsymbol{b}$. Note that the network is not homogeneous and therefore recent results on homogeneous networks do not apply (see Section 2). The network is convex because it is a sum of convex ReLU functions.

**Training Loss:**  To learn $f^*$ we aim to minimize the following hinge loss:

$$L(\boldsymbol{W}, \boldsymbol{b}) = \frac{1}{|\mathbb{S}|} \sum_{(\boldsymbol{x}, y) \in \mathbb{S}} \max\{0, 1 - yN(\boldsymbol{x}; \boldsymbol{W}, \boldsymbol{b})\} \tag{2}$$

We note that the above loss function is generally non-convex (even though the network $N$ is convex). For optimization we use Gradient Descent (GD) with a fixed learning rate $\eta$. We denote the initialization of GD by $\left(\boldsymbol{W}^{(0)}, \boldsymbol{b}^{(0)}\right)$ and the weights at iteration $t$ by $\left(\boldsymbol{W}^{(t)}, \boldsymbol{b}^{(t)}\right)$. If the iteration index is clear from context we omit it and use $(\boldsymbol{W}, \boldsymbol{b})$. See Section A for the gradient update.

In this work we will mainly focus on the population case. This corresponds to optimizing the loss in Eq. (2) with $\mathbb{S}_x = \mathcal{X}$.[4] Many works have studied it as a proxy to understand the performance in the empirical case (e.g., Brutzkus & Globerson, 2017; Daniely & Malach, 2020).

**Remark 3.1.** *The convex network considered here is a good test-bed for understanding the inductive bias of learning read-once DNFs with one-hidden layer NNs because: (1) It has the same expressive power as standard NNs for implementing Boolean functions (Section 4) (2) It outperforms standard NNs on learning read-once DNFs in our setting (Section 8) (3) Its analysis can be used to better understand the inductive bias of standard NNs (Section 8).*

## 4 EXPRESSIVE POWER

In this section we show that the network in Eq. (1) has the expressive power to implement *any* Boolean function over $\mathcal{X}$. Therefore, in terms of expressive power, the network is suitable for learning Boolean functions and has the same expressive power for implementing Boolean functions as a standard one-hidden layer NN.

**Theorem 4.1.** *Let $f : \mathcal{X} \to \mathcal{Y}$. Then, there exists $(\boldsymbol{W}, \boldsymbol{b})$ and a network $N$ in Eq. (1) with $r \leq 2^D$ neurons such that for all $\boldsymbol{x} \in \mathcal{X}$, $\text{sign}\left(N(\boldsymbol{x}; \boldsymbol{W}, \boldsymbol{b})\right) = \text{sign}\left(\sum_{i \in [r]} \sigma(\boldsymbol{w}_i \cdot \boldsymbol{x} + b_i) - 1\right) = f(\boldsymbol{x})$.*

*Proof.* Let $\mathbb{B}_+ = \{\boldsymbol{x} \in \mathcal{X} \mid f(\boldsymbol{x}) = 1\}$ and denote $\mathbb{B}_+ = \left\{\boldsymbol{x}_1, ..., \boldsymbol{x}_{|\mathbb{B}_+|}\right\}$. Define $r = |\mathbb{B}_+|$ and for each $i \in [r]$ define $\boldsymbol{w}_i = \boldsymbol{x}_i$ and $b_i = -D + 2$. Then $\forall \boldsymbol{x}_i \in \mathbb{B}_+$ it holds that $\sigma(\boldsymbol{w}_i \cdot \boldsymbol{x}_i + b_i) = 2$ and $\forall \boldsymbol{x} \neq \boldsymbol{x}_i$ it holds that $\sigma(\boldsymbol{w}_i \cdot \boldsymbol{x} + b_i) = 0$. Therefore $\forall \boldsymbol{x} \in \mathbb{B}_+$ we have $N(\boldsymbol{x}; \boldsymbol{W}, \boldsymbol{b}) = 1$ and for $\boldsymbol{x} \notin \mathbb{B}_+$ it holds that $N(\boldsymbol{x}; \boldsymbol{W}, \boldsymbol{b}) = -1$, from which the claim follows. $\qquad \square$

## 5 MULTIPLE GLOBAL MINIMA IN THE POPULATION SETTING

Assume that $\mathbb{S}_x = \mathcal{X}$. Then any global minimum of the loss in Eq. (2) implements the ground-truth function $f^*$. However, as we will show next, there are global minima that implement $f^*$ with drastically different properties. Understanding which global minimum GD converges to is important, because the population case is an approximation of the empirical case with sufficiently many training points. Thus a good understanding of the population case can have direct implications on our understanding of the empirical setting. Indeed, we show in Section 8 that using our understanding of the population case leads to a procedure that accurately reconstructs the ground-truth DNF from a network trained on practical-size training sets.

One network that globally minimizes the loss is the one whose neurons simply "memorize" all positive points, as the following Proposition states (proof follows from Theorem 4.1).

**Proposition 5.1.** *Assume $\mathbb{S}_x = \mathcal{X}$. Consider $(\boldsymbol{W}, \boldsymbol{b})$ with $r = |\mathbb{S}_p|$ s.t. for any $\boldsymbol{x} \in \mathbb{S}_p$ there exists $i \in [r]$ where $\boldsymbol{w}_i = \boldsymbol{x}$ and $b_i = -D + 2$. Then $(\boldsymbol{W}, \boldsymbol{b})$ is a global minimum of the loss in Eq. (2).*

---

[4]This follows since for the uniform distribution we have $\mathbb{E}_{\boldsymbol{x} \sim \mathcal{D}, y = f^*(\boldsymbol{x})}\left[\max\{0, 1 - yN(\boldsymbol{x}; \boldsymbol{W}, \boldsymbol{b})\}\right] = \frac{1}{|\mathcal{X}|} \sum_{\boldsymbol{x} \in \mathcal{X}, y = f^*(\boldsymbol{x})} \max\{0, 1 - yN(\boldsymbol{x}; \boldsymbol{W}, \boldsymbol{b})\}$.

We call the above minimum the *memorizing* solution. Intuitively, converging to this solution in the population case is undesirable, since this may imply memorization in the empirical setting, which can lead to wrong predictions on unobserved samples.

Next, we show a different global minimum which recovers the DNF formula explicitly in its neurons. We first need the following definitions.

**Definition 5.1.** *A neuron $i \in [r]$ is a* covering neuron *with respect to a DNF $f^*$, if there exists $n \in [K]$ and $\lambda_i > 0$ such that $\boldsymbol{w}_i = \lambda_i \boldsymbol{t}_n^*$. We refer to $\lambda_i$ as the* covering coefficient *of neuron $i$. We also refer to this as neuron $i$ covering the term $n$.*

**Definition 5.2.** $\boldsymbol{W}$ *covers $f^*$, if $\forall n \in [K]$ there exists $i \in [r]$ such that neuron $i$ covers term $n$.*

We can now define a DNF-recovery solution:

**Definition 5.3.** $\boldsymbol{W}$ *is a DNF-recovery solution if it covers $f^*$ and for neurons $i \in [r]$ that are not covering, it holds that $\boldsymbol{w}_i = 0$.*

For convenience, we will group all neurons that cover a specific term:

**Definition 5.4.** *Assume that $\boldsymbol{W}$ covers the DNF $f^*$. For each $n \in [K]$, we define the set $\mathbb{C}_n$ to be the set of all $i \in [r]$ such that neuron $i$ covers term $n$.*

Next, we show that any DNF-recovery solution is a global minimum under certain conditions.

**Proposition 5.2.** *Assume that $\mathbb{S}_x = \mathcal{X}$ and $\boldsymbol{W}$ is a DNF-recovery solution. Then $(\boldsymbol{W}, \boldsymbol{b})$ is a global minimum of the loss in Eq. (2) if the following holds. (1) For every covering neuron $i$ with covering coefficient $\lambda_i$ and which covers the term $n$, it holds that $b_i = \lambda_i(2 - \|\boldsymbol{t}_n^*\|_1)$. (2) For a neuron $i$ that is not covering, $b_i = 0$. (3) For all $n \in [K]$, $\sum_{i \in \mathbb{C}_n} \lambda_i \geq 1$.*

We note that the third condition ensures that for all $\boldsymbol{x} \in \mathbb{S}_p$ it holds that $N(\boldsymbol{c}; \boldsymbol{W}, \boldsymbol{b}) \geq 1$. We need this condition to ensure global optimality. The proof is provided in Section B. Intuitively, converging to this global minimum is desirable, because it learns good representations of the data: the terms of the ground-truth read-once DNF $f^*$. Thus, given this global minimum, the read-once DNF can be easily reconstructed from the network weights, which can be useful for interpretability. Intuitively, since a DNF-recovery solution is equivalent to a network with a small number of neurons, converging to it in the population case may suggest good sample complexity in the empirical case. Indeed, we show empirically in Section 8 that the convex network has good sample complexity for learning read-once DNFs in our setting. The remaining question is which global minimum GD converges to.

# 6   CONVERGENCE ANALYSIS WITH A COMPUTER ASSISTED PROOF

In this section we characterize the global minimum that GD converges to in the population setting. Answering this question via theoretical analysis is extremely challenging due to the non-convexity and complex dynamics of GD. Therefore, to tackle this problem, we opt for a computer assisted proof. Using a computer assisted proof we show that for low dimensional DNFs, GD converges to a solution which is similar to the DNF-recovery solution in the following sense: after a simple pruning and reconstruction procedure, the DNF-recovery solution can be obtained from the global minimum of GD (see Theorem 6.1). We use a unique property of our setting (see Lemma 6.1) that allows us to perform calculations in integers and avoid floating point errors. In Section 7, we provide further theoretical results on the dynamics of GD that corroborate our findings.

## 6.1   SETUP

We next provide details on the setup of the computer assisted proof. See Section I for more details.
**Network and Algorithm:** We consider the network in Eq. (1) with $r = 2^D$ and GD with initialization $\boldsymbol{W}^{(0)}$ corresponding to all possible vectors in $\{\pm\epsilon\}^D$ and $b^{(0)} = 0$.[5] We execute the proof for $D \leq 15$. We use small values of $\epsilon$ to converge to global minima that are similar to Figure 1c. For

---

[5]We chose a single symmetric initialization of all binary vectors with entries in $\{\pm\epsilon\}$ to serve as a representative of initializations used in practice. Indeed, we show in Section 8 that the conclusions of Theorem 6.1, empirically hold for Gaussian initializations.

large initializations, GD has a different inductive bias (see Section 8). Details on values of $\epsilon$ and learning rate are provided in Section I.

**DNFs:** We define a *balanced* DNF to be a read-once DNF such that for all $i \neq j \in [K]$, $|\mathbb{A}_i| = |\mathbb{A}_j|$. For the proof, we consider all balanced read-once DNFs $f^*$ with input dimension $4 \leq D \leq 12$ where each term is of size at least two.[6] We denote the latter set of DNFs by $\mathbb{F}$.

**Pruning and Reconstruction:** Figure 1c shows an example of the global minimum that GD converges to. Most of the neurons cover a term, but not all. Except for the non-covering neurons, the global minimum looks very similar to a DNF-recovery solution. Based on this observation, we will devise a pruning and rounding procedure that when applied to the global minimum of GD, will provably return a DNF-recovery solution for the cases we consider in the computer assisted proof. Section 8 shows that this procedure also works empirically for various other settings. Next we define the pruning procedure we will use.

**Definition 6.1.** *For $0 \leq \gamma \leq 1$ and a network $N$ with parameters $(\boldsymbol{W}, \boldsymbol{b})$, define $\mathbb{Q}_\gamma(\boldsymbol{W}) = \{i \in [r] \mid \|\boldsymbol{w}_i\|_\infty > \gamma M_\infty(\boldsymbol{W})\}$ where $M_\infty(\boldsymbol{W}) = \max\limits_{i \in [r]} \{\|\boldsymbol{w}_i\|_\infty\}$.*

The pruning procedure removes all neurons that have $l_\infty$ norm which is less than $\gamma M_\infty(\boldsymbol{W})$. Next we define the reconstruction procedure.

**Definition 6.2.** *For $0 \leq \beta \leq 1$, a $\beta$-reconstruction of weight matrix $\boldsymbol{W}$ is $R_\beta(\boldsymbol{W}) \in \{0,1\}^{rD}$ such that for all $i \in [r]$ and $j \in [D]$, $R_\beta(\boldsymbol{W})_{ij} = 1$ if $w_{ij} > \beta \|\boldsymbol{w}_i\|_\infty$, or $R_\beta(\boldsymbol{W})_{ij} = 0$ otherwise.*

Figure 2 shows an example of the pruning and reconstruction procedures. We use the values of $0.4 \leq \beta \leq 0.9$ and $0.4 \leq \gamma \leq 0.9$ in our proof. Further details are given in Section I.

## 6.2 MAIN RESULT

We can now state our main result, which states that for the set of DNFs $\mathbb{F}$ it holds that GD in the setup of Section 6.1 followed by rounding will converge to a DNF-recovery solution. See Figure 2 for an example of this result.

**Theorem 6.1.** *For all $f^* \in \mathbb{F}$, parameters $\eta, \beta, \gamma, \epsilon$ as described in Section I, and $\mathbb{S}_x = \mathcal{X}$, GD converges to a global minimum $(\boldsymbol{W}, \boldsymbol{b})$ such that $R_\beta(\mathbb{Q}_\gamma(\boldsymbol{W}))$ is a DNF-recovery solution.*

We next provide a sketch of the proof. The main challenge is of course simulating the GD updates, while avoiding floating point errors. We use a key observation that in our setting, we can perform equivalent calculations of the dynamics in *integers*. This follows since in our binary input setting, the network dynamics can be calculated with rational numbers. Then, by scaling several parameters, we can perform equivalent calculations in integers. The scaling procedure is defined below.

**Definition 6.3.** *Given $\alpha > 0$, weight $\boldsymbol{W}^{(0)}$ and learning rate $\eta$, the $\alpha$-GD algorithm is defined as follows. Initialize $\boldsymbol{U}^{(t)} = \alpha \boldsymbol{W}^{(0)}$, $\boldsymbol{c}^{(0)} = 0$. Then run GD with constant learning rate $\eta_\alpha = \eta\alpha$ with loss $L_\alpha(\boldsymbol{U}, \boldsymbol{c}) = \frac{1}{\mathbb{S}} \sum\limits_{(\boldsymbol{x},y) \in \mathbb{S}} \max\{0, \alpha - yN_\alpha(\boldsymbol{x}; \boldsymbol{U}, \boldsymbol{c})\}$, where $N_\alpha(\boldsymbol{x}; \boldsymbol{U}, \boldsymbol{c}) = \sum\limits_{i \in [r]} \sigma(\boldsymbol{u}_i \cdot \boldsymbol{x} + c_i) - \alpha$.*

Since the inputs are integers, by taking the learning rate and $\epsilon$ to be rational numbers and $\alpha$ to be a sufficiently large integer, $\alpha$-GD can perform all calculations in integers. Next, we show that $\alpha$-GD performs the same calculations as GD up to a scaling factor. Therefore, we can run $\alpha$-GD to simulate GD without floating point errors.

**Lemma 6.1.** *Assume we run GD with initialization $(\boldsymbol{W}^{(0)}, \boldsymbol{b}^{(0)})$ where $\boldsymbol{b}^{(0)} = 0$ and constant learning rate $\eta$ to optimize the loss in Eq. (2) and assume we run $\alpha$-GD with paramaters $\boldsymbol{W}^{(0)}$ and learning rate $\eta$. Then it holds that $\alpha\boldsymbol{W}^t = \boldsymbol{U}^t$, $\alpha\boldsymbol{b}^t = \boldsymbol{c}^t$ and $\alpha L(\boldsymbol{W}^t, \boldsymbol{b}^t) = L_\alpha(\boldsymbol{U}^t, \boldsymbol{c}^t)$.*

We prove this lemma by induction on $t$ (see Section E). For the computer assisted proof, we run $\alpha$-GD for each $f^* \in \mathbb{F}$ and data $\mathbb{S}_x = \mathcal{X}$ labeled by $f^*$, with suitable $\alpha$, rational $\epsilon$ and $\eta$ such that all calculations are performed in integers. We note that the pruning and reconstruction procedures can be performed with integers as well. Further details are given in Section I.

---

[6] We focus on balanced DNFs because while the symmetric initialization $\boldsymbol{W}^{(0)}$ described above is a good test-bed for understanding the inductive bias for balanced DNFs, it is not so for unbalanced DNFs. Indeed, we observe empirically that GD converges in some instances to spurious local minima in this case. However, we will empirically show in Section 8 that this is not the case for Gaussian initialization and that for this initialization we can successfully reconstruct DNFs using the method described in this section.

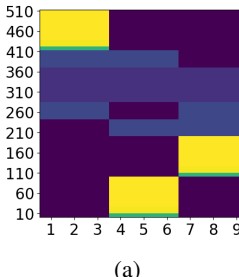 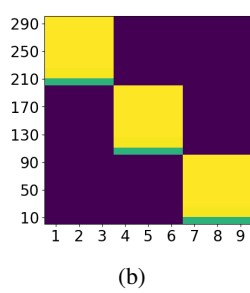 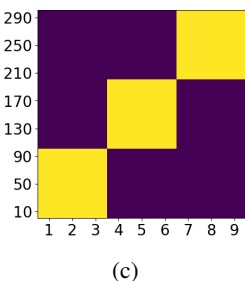

(a)               (b)               (c)

Figure 2: Illustration of pruning and reconstruction for the read-once DNF: $(x_1 \wedge x_2 \wedge x_3) \vee (x_4 \wedge x_5 \wedge x_6) \vee (x_7 \wedge x_8 \wedge x_9)$. (a) The global minimum that GD converges to. (b) Global minimum after pruning (Definition 6.1). (c) Global minimum after pruning followed by $\beta$-reconstruction (Definition 6.2).

## 7 THEORETICAL INSIGHTS

In this section we give two theoretical results shedding further light on the inductive bias of GD.

**Preservation of Term Alignment:** Assume the symmetric initialization in the computer assisted proof. In the next theorem we show that if at some iteration $T$ of GD, all weight entries in a term are equal, then this will continue to hold for all $t > T$. This provides further evidence that GD is biased towards term alignment.

**Theorem 7.1.** *Assume GD is initialized with $(\boldsymbol{W}^0, \boldsymbol{b}^0)$ as described in Section 6 and $\mathbb{S}_x = \mathcal{X}$. Assume that there exists $T \geq 0$, $i \in [r]$ and $n \in [K]$ such that for all $j_1, j_2 \in \mathbb{A}_n$, $w_{ij_1}^{(T)} = w_{ij_2}^{(T)}$. Then for all $t > T$, $j_1, j_2 \in \mathbb{A}_n$ it holds that $w_{ij_1}^{(t)} = w_{ij_2}^{(t)}$.*

The proof idea is to exploit the symmetry of the initialization and population setting to show that the GD update is constrained to preserve term alignment. The proof is given in Section F.

**Norm Minimization implies DNF-Recovery:** Recent works have highlighted interesting connections between gradient methods and norm minimization (Lyu & Li, 2020; Nacson et al., 2019). The optimization problem is to minimize the norm of the model weights subject to the constraint $yN(\boldsymbol{x}; \boldsymbol{W}, \boldsymbol{b}) \geq 1$ for all $(\boldsymbol{x}, y)$ in the training data. In Lyu & Li (2020) it was shown that under some conditions GD will converge to a KKT point of this problem. We next show a surprising result in our context: the global optimum of this min-norm problem is a DNF-recovery solution. This means that if GD converges to the optimal KKT point, it will find a DNF-recovery solution. [7]

**Theorem 7.2.** *Consider the margin-maximization optimization problem:*

$$\begin{aligned} \min \quad & \textstyle\sum_{i \in [r]} \|(\boldsymbol{w}_i, b_i)\|_2^2 \\ s.t. \quad & yN(\boldsymbol{x}; \boldsymbol{W}, \boldsymbol{b}) \geq 1 \,, \ \forall (\boldsymbol{x}, y) \in \mathbb{S} \end{aligned} \tag{3}$$

*Then, for any solution $(\boldsymbol{W}^*, \boldsymbol{b}^*)$ of Eq. (3), $\boldsymbol{W}^*$ is a DNF-recovery solution.*

The proof is technical and given in Section G. The key idea is to show an upper bound on the bias of any global minimum and that the bound is tight for a minimum $l_2$ norm solution. Then, using this fact together with the optimality of $(\boldsymbol{W}^*, \boldsymbol{b}^*)$ the theorem can be proved.

## 8 EMPIRICAL RESULTS

In this section we perform numerous experiments that support our analysis and show that our conclusions hold in different settings. For each experiment we show a sample of the empirical results due to space constraints. Further details and results are provided in the supplementary.

**Comparing Convex and Standard Networks:** Our analysis focused on a convex network. Here we compare it to a standard 2-layer network with trainable output layer. Figure 3a reports results,

---

[7]For the hinge loss, we do not expect GD to converge exactly to the min norm solution. Indeed, Figure 2a from the computer assisted proof shows that GD converges to a solution which is not a DNF recovery solution.

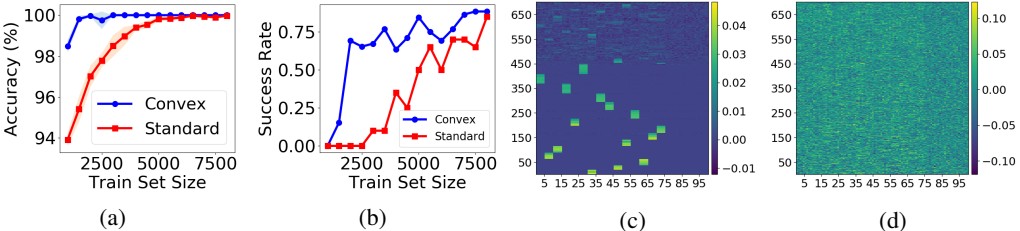

Figure 3: Experiments for learning read-once DNFs. (a) Test performance of convex and standard two-layer networks for $D = 9$. (b) Reconstruction success rate for convex and standard two-layer networks, for $D = 27$. (c) Model learned by a convex network for DNF with $D = 100$ and small initialization (d) Same setting as (c) with large initialization.

showing that the convex network outperforms the standard one for a DNF with $D = 9$. This shows that the convex network is a good model for studying inductive bias in our setting.

**Reconstruction of DNFs in Other Settings:** Theorem 6.1 holds under the assumption of low dimensional DNFs, a convex network, population case, symmetric initialization and balanced DNFs. Here we show that the reconstruction procedure in the theorem can be applied in broader settings. Specifically, we consider a setting with Gaussian initialization, finite data and unbalanced high dimensional DNFs for $D = 27$. Figure 3b shows the reconstruction success rate for both networks. For each training set size, the recovery is performed for different initializations and training sets. Note that for standard NNs we slightly modified the recovery procedure to accommodate for these networks. In both cases, we see that for at least a moderate training set size, the recovery procedure has a high success rate (recall this is for *perfect* recovery). We note that for $D > 27$ the recovery for standard neural networks did not work well. However, for the convex network even for $D = 100$, the reconstruction worked well (see Section H). Figure 3c shows an example of a global minimum of GD for $D = 100$, where the DNF recovery was successful.

**Large Initialization:** Our results in Section 6 required a small initialization scale. However, several works have shown that the scale of the initialization has a significant effect on the inductive bias (Woodworth et al., 2019; Chizat et al., 2019). What is the inductive bias in the case of large initialization in our setting? Figure 3d shows the neurons of the *global minimum* that GD converges to for large initialization and the setting of Figure 3c. We see that neurons are not aligned with terms and have a very different inductive bias. Indeed, this model also overfits with $67\%$ test accuracy, compared to $100\%$ test accuracy for the small-scale initialization model in Figure 3c.

**Experiments on Tabular Datasets:** The fact that SGD recovers simple Boolean formulas is very attractive in the context of interpretability. We showed that we can reconstruct DNFs under certain idealized assumptions (e.g., uniform distribution, read-once). However, our reconstruction method might produce meaningless reconstructions on datasets which are not uniform nor labeled with a read-once DNF. We tested our reconstruction method on three tabular UCI datasets kr-vs-kp, diabetes and Splice (Dua & Graff, 2017). Learning with our convex network resulted in test accuracies of 93%, 96% and 96% on these datasets, respectively. Our reconstruction method obtained a small DNF (3 terms of size less than 3) on kr-vs-kp with test accuracy 83%. For diabetes, the reconstruction method returned a large DNF (more than 10 terms) with test accuracy $93\%$. On Splice we got a 2-term DNF of sizes 2 and 3 with $95\%$ test accuracy. The latter is a very compact DNF with very small loss in accuracy, illustrating the potential of recovery on interpretability.

## 9 CONCLUSIONS

Understanding the inductive bias of neural networks for learning DNFs is an important challenge. In this work we mainly focused on learning read-once DNFs under the uniform distribution with a convex network. We provided theoretical results, computer assisted proofs and experiments, all of which suggest that GD is biased towards unique global minima that recover the terms of the DNF. Using our analysis we derived a DNF reconstruction method and showed that it works in broader settings that include standard two-layer networks and tabular datasets.

Our work opens up many interesting directions for future work. For example, it would be interesting to understand if DNF recovery is possible for other distributions and DNFs that are not read-once. Another interesting direction is to understand the sample complexity of neural networks for learning DNFs and how it relates to DNF-recovery. Finally, it will be interesting to understand how learning dynamics in neural nets are related to other algorithms for learning DNFs.

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

## A   GRADIENT UPDATE

The following definition will be used in order to simplify the gradient updates of GD:

**Definition A.1.** *Given* $(\boldsymbol{W}^{(t)}, \boldsymbol{b}^{(t)})$*, for each neuron* $i \in [r]$ *define:*

$$\mathbb{G}_i^{(t)} = \{(\boldsymbol{x}, y) \in \mathbb{S} \mid 1 - yN(\boldsymbol{x}; \boldsymbol{W}^{(t)}, \boldsymbol{b}^{(t)}) > 0 \ \wedge \ \boldsymbol{w}_i^{(t)} \cdot \boldsymbol{x} + b_i^{(t)} > 0\} \tag{4}$$

The set $\mathbb{G}_i^{(t)}$ consists of all samples that are included in the gradient update for neuron $i$ at time $t$. Using this definition, the update rule of GD for neuron $i$ at time $t$ is given by:

$$\boldsymbol{w}_i^{(t)} = \boldsymbol{w}_i^{(t-1)} + \frac{\eta}{|\mathbb{S}|} \sum_{(\boldsymbol{x}, y) \in \mathbb{G}_i^{(t-1)}} y\boldsymbol{x} \quad ; \quad b_i^{(t)} = b_i^{(t-1)} + \frac{\eta}{|\mathbb{S}|} \sum_{(\boldsymbol{x}, y) \in \mathbb{G}_i^{(t-1)}} y \tag{5}$$

## B   PROOF OF PROPOSITION 5.2

To prove the result we show necessary properties of global minima when $\mathbb{S}_x = \mathcal{X}$ that will also be useful in the proof of Theorem 7.2. The proof of Proposition 5.2 follows directly from Lemma B.2 below.

### B.1   A SIMPLE PROPERTY OF GLOBAL MINIMA

Recall the definition $\mathbb{S}_p = \{\boldsymbol{x} \mid (\boldsymbol{x}, 1) \in \mathbb{S}\}$. We also define $\mathbb{S}_n = \{\boldsymbol{x} \mid (\boldsymbol{x}, -1) \in \mathbb{S}\}$. We first need the following definitions.

**Definition B.1.** *We say that* $(\boldsymbol{W}, \boldsymbol{b})$ *satisfies the* $MIN_+$ *property if* $\forall \boldsymbol{x} \in \mathbb{S}_p$ *there exists* $\mathbb{I} \subseteq [r]$ *such that* $\sum_{i \in \mathbb{I}} \boldsymbol{w}_i \cdot \boldsymbol{x} + b_i \geq 2$.

**Definition B.2.** *We say that* $(\boldsymbol{W}, \boldsymbol{b})$ *satisfies the* $MIN_-$ *property if* $\forall \boldsymbol{x} \in \mathbb{S}_n$, $\forall i \in [r]$ $\boldsymbol{w}_i \cdot \boldsymbol{x} + b_i \leq 0$.

The following property of a global minimum follows directly by the definition of the network in Eq. (1) and the loss function in Eq. (2). The proof is given in Section C.

**Lemma B.1.** *Assume that* $\mathbb{S}_x = \mathcal{X}$. *Then* $(\boldsymbol{W}, \boldsymbol{b})$ *is a global minimum of Eq. (2) if and only if* $(\boldsymbol{W}, \boldsymbol{b})$ *satisfies* $MIN_+$ *and* $MIN_-$.

We note that one direct consequence of Lemma B.1 is that if a negative $\boldsymbol{x} \in \mathbb{S}_n$ is activated by a neuron $i \in [r]$, i.e., $\boldsymbol{w}_i \cdot \boldsymbol{x} + b_i > 0$, then $(\boldsymbol{x}, y) \in \mathbb{G}_i$. We will use this fact in later sections.

### B.2   THE BIAS THRESHOLD

In this section we show that when $\mathbb{S}_x = \mathcal{X}$, the bias of any neuron in a global minimum is upper bounded by a certain value which we call the *bias threshold*. This upper bound will be useful in the proof of Theorem 7.2.

**Definition B.3.** *For each term* $n \in [K]$ *of* $f^*$ *and* $i \in [r]$ *define* $V_n(\boldsymbol{w}_i) = \max\{\min_{j \in \mathbb{A}_n}\{w_{ij}\}, 0\}$.

**Definition B.4.** *The bias threshold for a weight* $\boldsymbol{w}$ *is* $BT(\boldsymbol{w}) = -\|\boldsymbol{w}\|_1 + 2\sum_{n \in [K]} V_n(\boldsymbol{w})$.

**Lemma B.2.** *Assume that* $\mathbb{S}_x = \mathcal{X}$. *If* $(\boldsymbol{W}, \boldsymbol{b})$ *satisfies that* $\forall i \in [r]$ $b_i \leq BT(\boldsymbol{w}_i)$ *and satisfies the* $MIN_+$ *property. Then it is a global minimum of the loss in Eq. (2).*

The proof is given in Section D. The idea is to find for each $i \in [r]$ a negative point $\hat{\boldsymbol{x}}_i$ such that $\boldsymbol{w}_i \cdot \hat{\boldsymbol{x}}_i = BT(\boldsymbol{w}_i)$ and that for any other negative point $\boldsymbol{x}_i$, $\boldsymbol{w}_i \cdot \boldsymbol{x}_i \leq \boldsymbol{w}_i \cdot \hat{\boldsymbol{x}}_i$. From this, we show that the $MIN_-$ property is satisfied if and only if $\forall i \in [r]$, $b_i \leq BT(\boldsymbol{w}_i)$. Then, using Lemma B.1 we complete the proof.

## C   PROOF OF LEMMA B.1

*Proof.* If $(\boldsymbol{W}, \boldsymbol{b})$ is a global minimum, then for all $(\boldsymbol{x}, y) \in \mathbb{S}$, $yN(\boldsymbol{x}; \boldsymbol{W}, \boldsymbol{b}) \geq 1$. Therefore, if $y = 1$, $\sum_{i \in [r]} \sigma(\boldsymbol{w}_i \cdot \boldsymbol{x} + b_i) \geq 2$. Thus, there exists $\mathbb{I} \subseteq [r]$ such that $\sum_{i \in \mathbb{I}} \boldsymbol{w}_i \cdot \boldsymbol{x} + b_i \geq 2$. If $y = -1$,

then $\sum_{i \in [r]} \sigma(\boldsymbol{w}_i \cdot \boldsymbol{x} + b_i) \le 0$ and therefore for all $i \in [r]$, $\boldsymbol{w}_i \cdot \boldsymbol{x} + b_i \le 0$. The other direction follows similarly. $\qquad\square$

## D  PROOF OF LEMMA B.2

**Definition D.1.** *We define the set of indices which are not active in any term as the noisy indices and denote them by $\mathbb{A}_{K+1} = [D] \setminus \cup_{n \in [K]} \mathbb{A}_n$.*

The lemma follows directly from Lemma B.1 and the following lemma.

**Lemma D.1.** *Given a neuron $(\boldsymbol{w}, b)$, there is negative sample $\boldsymbol{x} \in \mathbb{S}_n$ for which $\boldsymbol{w} \cdot \boldsymbol{x} + b > 0$ if and only if $b > BT(\boldsymbol{w})$.*

*Proof.* Given neuron $(\boldsymbol{w}, b)$, we define the minimum index of a term $n \in [K]$ as $J_n = \arg\min_{j \in \mathbb{A}_n} \{w_j\}$. Consider a sample $\hat{\boldsymbol{x}} \in \mathbb{S}_x$ where for each $j \in [D]$:

$$\hat{x}_j = \begin{cases} -\operatorname{sign}(w_j) & \exists n \in [K]: \ V_n(\boldsymbol{w}) > 0 \ \wedge \ j = J_n \\ \operatorname{sign}(w_j) & \textbf{otherwise} \end{cases} \tag{6}$$

For a term $n \in [K]$, if $V_n(\boldsymbol{w}) > 0$ then $\hat{x}_{J_n} = -\operatorname{sign}(\boldsymbol{w}_{J_n}) = -1$, otherwise $V_n(\boldsymbol{w}) = 0$ and there exists $j \in \mathbb{A}_n$ such that $w_j < 0$, i.e., $\boldsymbol{x}_j = \operatorname{sign}(w_j) = -1$. In any case, $\boldsymbol{x} \cdot \boldsymbol{t}_n^* < |\mathbb{A}_n|$. Therefore, the label of this sample is negative and denote it by $\hat{y} = -1$.

Now, we have the following:

$$
\begin{aligned}
\boldsymbol{w} \cdot \hat{\boldsymbol{x}} &= \sum_{j \in [D]} w_j \cdot \hat{x}_j = \sum_{n \in [K+1]} \left[ \sum_{j \in \mathbb{A}_n} w_j \cdot \hat{x}_j \right] \\
&= \sum_{n \in [K] \text{ and } V_n(\boldsymbol{w}) > 0} \left[ \sum_{j \in \mathbb{A}_n \setminus \{J_n\}} w_j \cdot \operatorname{sign}(w_j) - w_{J_n} \cdot \operatorname{sign}(w_{J_n}) \right] \\
&\quad + \sum_{n \in [K] \text{ and } V_n(\boldsymbol{w}) = 0} \left[ \sum_{j \in \mathbb{A}_n} w_j \cdot \operatorname{sign}(w_j) \right] + \sum_{j \in \mathbb{A}_{K+1}} w_j \cdot \hat{x}_j \\
&= \sum_{n \in [K] \text{ and } V_n(\boldsymbol{w}) > 0} \left[ \sum_{j \in \mathbb{A}_n \setminus \{J_n\}} |w_j| - |w_{J_n}| \right] + \sum_{n \in [K] \text{ and } V_n(\boldsymbol{w}) = 0} \left[ \sum_{j \in AI_n} |w_j| \right] + \sum_{j \in \mathbb{A}_{K+1}} |w_j| \\
&= \sum_{n \in [K] \text{ and } V_n(\boldsymbol{w}) > 0} \left[ \sum_{j \in \mathbb{A}_n} |w_j| - 2V_n(\boldsymbol{w}) \right] + \sum_{n \in [K] \text{ and } V_n(\boldsymbol{w}) = 0} \left[ \sum_{j \in AI_n} |w_j| - 2V_n(\boldsymbol{w}) \right] + \sum_{j \in \mathbb{A}_{K+1}} |w_j| \\
&= \sum_{n \in [K]} \left[ \sum_{j \in AI_n} |w_j| - 2V_n(\boldsymbol{w}) \right] + \sum_{j \in \mathbb{A}_{K+1}} |w_j| = \sum_{j \in [D]} |w_j| - 2 \sum_{n \in [K]} V_n(\boldsymbol{w}) \\
&= \|\boldsymbol{w}\|_1 - 2 \sum_{n \in [K]} V_n(\boldsymbol{w}) = -BT(\boldsymbol{w})
\end{aligned}
\tag{7}
$$

For the first direction, if $b > BT(\boldsymbol{w})$ then $\boldsymbol{w} \cdot \hat{\boldsymbol{x}} + b = BT(\boldsymbol{w}) + b > 0$, as desired.

In the second direction, assume that there is a negative sample $\boldsymbol{x} \in \mathbb{S}_n$ such that $\boldsymbol{w} \cdot \boldsymbol{x} + b > 0$. We will show that $\boldsymbol{x} \cdot \boldsymbol{w} \le \hat{\boldsymbol{x}} \cdot \boldsymbol{w}$. Every term $n \in [K]$ satisfies:

$$\forall j \in \mathbb{A}_n \setminus \{J_n\} \quad x_j w_j \le |w_j| = \operatorname{sign}(w_j) w_j = \hat{x}_j w_j \tag{8}$$

If $V_n(\boldsymbol{w}) = 0$, then the index $J_n$ also satisfies Eq. (8), by the definition of $\hat{\boldsymbol{x}}$. Otherwise $V_n(\boldsymbol{w}) > 0$ and we know that there exists $j' \in \mathbb{A}_n$ such that $x_{j'} = -1$ (otherwise the sample's label is not negative), and $w_{J_n}, w_{j'} > 0$. If $J_n = j'$, then $x_j w_j = \hat{x}_j w_j$. Otherwise, the following holds:

$$x_{j'} w_{j'} + x_{J_n} w_{J_n} \le -w_{j'} + w_{J_n} \le 0 \le w_{j'} - w_{J_n} \le \hat{x}_{j'} w_{j'} + \hat{x}_{J_n} w_{J_n} \tag{9}$$

Note that every every index in $A_{K+1}$ satisfies Eq. (8) as well. Therefore, $\boldsymbol{x} \cdot \boldsymbol{w} \leq \hat{\boldsymbol{x}} \cdot \boldsymbol{w}$. We can conclude that:

$$0 < \boldsymbol{x} \cdot \boldsymbol{w} + b \leq \hat{\boldsymbol{x}} \cdot \boldsymbol{w} + b = -BT(\boldsymbol{w}) + b \tag{10}$$

which implies that $b > BT(\boldsymbol{w})$ as desired. $\qquad\square$

## E  PROOF OF LEMMA 6.1

*Proof.* We wish to prove that the $\alpha$-GD optimization procedure is equivalent to the original GD, up to scaling. We begin with some definitions.

**Definition E.1.** *Given* $(\boldsymbol{U}^{(t)}, \boldsymbol{c}^{(t)})$, *for a neuron* $i \in [r]$ *is defined:*

$$\mathbb{H}_i^{(t)} = \{(\boldsymbol{x}, y) \in \mathbb{S} \mid 1 - yN_\alpha(\boldsymbol{x}; \boldsymbol{U}^{(t)}, \boldsymbol{c}^{(t)}) > 0 \ \wedge \ \boldsymbol{u}_i^{(t)} \cdot \boldsymbol{x} + c_i^{(t)} > 0\} \tag{11}$$

The update rule of $\alpha$-GD for $(\boldsymbol{U}^{(t)}, \boldsymbol{c}^{(t)})$ can be simplified as follows:

$$\boldsymbol{u}_i^{(t)} = \boldsymbol{u}_i^{(t-1)} + \frac{\eta}{|\mathbb{S}|} \sum_{(\boldsymbol{x}, y) \in \mathbb{H}_i^{(t-1)}} y\boldsymbol{x} \quad ; \quad c_i^{(t)} = c_i^{(t-1)} + \frac{\eta}{|\mathbb{S}|} \sum_{(\boldsymbol{x}, y) \in \mathbb{H}_i^{(t-1)}} y \tag{12}$$

We prove the claim by induction on $t$. For $t = 0$ by the definition of the initialization we have:

$$\boldsymbol{U}^{(0)} = [\pm\alpha\epsilon]^D = \alpha[\pm\epsilon]^D = \alpha\boldsymbol{W}^{(0)} \quad ; \quad \boldsymbol{c}^{(0)} = [0]^D = \alpha\boldsymbol{b}^{(0)} \tag{13}$$

The network satisfies:

$$N_\alpha(\boldsymbol{x}; \boldsymbol{U}^{(0)}, \boldsymbol{c}^{(0)}) = \sum_{i \in [r]} \sigma(\boldsymbol{u}_i^{(0)} \cdot \boldsymbol{x} + c_i^{(0)}) - \alpha = \sum_{i \in [r]} \sigma(\alpha\boldsymbol{w}_i^{(0)} \cdot \boldsymbol{x} + \alpha b_i^{(0)}) - \alpha$$

$$= \sum_{i \in [r]} \alpha\sigma(\boldsymbol{w}_i^{(0)} \cdot \boldsymbol{x} + b_i^{(0)}) - \alpha = \alpha\left(\sum_{i \in [r]} \sigma(\boldsymbol{w}_i^{(0)} \cdot \boldsymbol{x} + b_i^{(0)})\right) \tag{14}$$

$$= \alpha N(\boldsymbol{x}; \boldsymbol{W}^{(0)}, \boldsymbol{b}^{(0)})$$

and we can conclude that:

$$L_\alpha(\boldsymbol{U}^{(0)}, \boldsymbol{c}^{(0)}) = \frac{1}{\mathbb{S}} \sum_{(\boldsymbol{x}, y) \in \mathbb{S}} \max\{0, \alpha - yN_\alpha(\boldsymbol{x}; \boldsymbol{U}^{(0)}, \boldsymbol{c}^{(0)})\}$$

$$= \frac{1}{\mathbb{S}} \sum_{(\boldsymbol{x}, y) \in \mathbb{S}} \max\{0, \alpha - y\alpha N(\boldsymbol{x}; \boldsymbol{W}^{(0)}, \boldsymbol{b}^{(0)})\} \tag{15}$$

$$= \alpha\frac{1}{\mathbb{S}} \sum_{(\boldsymbol{x}, y) \in \mathbb{S}} \max\{0, 1 - yN(\boldsymbol{x}; \boldsymbol{W}^{(0)}, \boldsymbol{b}^{(0)})\}$$

$$= \alpha L(\boldsymbol{W}^{(0)}, \boldsymbol{b}^{(0)})$$

For the induction step, we assume correctness of the claim for $t - 1$, and prove it for $t$. First, we can see that by the induction assumption for $\boldsymbol{x} \in \mathbb{S}_x$ and $i \in [r]$ we have that $\boldsymbol{u}_i^{(t-1)} \cdot \boldsymbol{x} + c_i^{(t-1)} > 0$ if and only if $\boldsymbol{w}_i^{(t-1)} \cdot \boldsymbol{x} + b_i^{(t-1)} > 0$. In addition, we have that $\alpha - yN_\alpha(\boldsymbol{x}; \boldsymbol{U}^{(t-1)}, \boldsymbol{c}^{(t-1)}) > 0$ if and only if $1 - yN(\boldsymbol{x}; \boldsymbol{W}^{(t-1)}, \boldsymbol{b}^{(t-1)})\} > 0$.

By Definition A.1 and Definition E.1 and the induction hypothesis, for every $i \in [r]$ it holds that $\mathbb{G}_i^{(t-1)} = \mathbb{H}_i^{(t-1)}$. Then, by the induction hypothesis, the following holds:

$$\boldsymbol{u}_i^{(t)} = \boldsymbol{u}_i^{(t-1)} + \frac{\alpha\eta}{|\mathbb{S}|} \sum_{(\boldsymbol{x}, y) \in \mathbb{H}_i^{(t-1)}} y\boldsymbol{x} = \alpha\boldsymbol{w}_i^{(t-1)} + \frac{\alpha\eta}{|\mathbb{S}|} \sum_{(\boldsymbol{x}, y) \in \mathbb{G}_i^{(t-1)}} y\boldsymbol{x} = \alpha\boldsymbol{w}_i^{(t)} \tag{16}$$

$$c_i^{(t)} = c_i^{(t-1)} + \frac{\alpha\eta}{|\mathbb{S}|} \sum_{(\boldsymbol{x}, y) \in \mathbb{H}_i^{(t-1)}} y = \alpha b_i^{(t-1)} + \frac{\alpha\eta}{|\mathbb{S}|} \sum_{(\boldsymbol{x}, y) \in \mathbb{G}_i^{(t-1)}} y = \alpha b_i^{(t)}$$

as required for the first condition.

To prove that $N_\alpha(\boldsymbol{x}; \boldsymbol{U}^{(t)}, \boldsymbol{c}^{(t)}) = \alpha N(\boldsymbol{x}; \boldsymbol{W}^{(t)}, \boldsymbol{b}^{(t)})$ and $\alpha L(\boldsymbol{W}^t, \boldsymbol{b}^t) = L_\alpha(\boldsymbol{U}^t, \boldsymbol{c}^t)$, we can repeat the process of Eq. (14) and Eq. (15) with iteration $t$ instead of 0. $\qquad\square$

# F    PROOFS OF THEOREM 7.1 (PRESERVATION OF TERM ALIGNMENT)

We use Definition D.1 where $\mathbb{A}_{K+1} = [D]\backslash \cup_{n\in[K]} \mathbb{A}_n$. We first prove a symmetry lemma in Section F.1 and then prove the theorem in Section F.2.

## F.1    SYMMETRY LEMMA

**Definition F.1.** *We define a reordering of $\mathbb{A}_1,\dots,\mathbb{A}_{K+1}$ as the set $\mathcal{R} = \{\pi_1,\dots,\pi_{K+1}\}$, where each $\pi_i$ is a permutation of elements in $\mathbb{A}_i$ for all $1 \le i \le K+1$.*

**Definition F.2.** *Given a reordering $\mathcal{R}$, we define for every sample $(\boldsymbol{x},y) \in \mathbb{S}$ a pair $(\boldsymbol{x}^\pi, y^\pi)$ such that:*

$$\forall n \in [K+1] \ \forall j \in \mathbb{A}_n \quad x^\pi_{\pi_n[j]} = x_j \tag{17}$$

*and $y^\pi = y$.*

Since the label of a sample is invariant to a reordering, it holds that $(\boldsymbol{x}^\pi, y^\pi) \in \mathbb{S}$.

**Lemma F.1.** *Given $(\boldsymbol{W}^{(0)}, \boldsymbol{b}^{(0)})$, a reordering $\mathcal{R}$ and step $t \ge 0$, every $\boldsymbol{x} \in \mathbb{S}_x$ satisfies:*

$$N(\boldsymbol{x}; \boldsymbol{W}^{(t)}, \boldsymbol{b}^{(t)}) = N(\boldsymbol{x}^\pi; \boldsymbol{W}^{(t)}, \boldsymbol{b}^{(t)}) \tag{18}$$

*Proof.* Given $(\boldsymbol{W}^{(0)}, \boldsymbol{b}^{(0)})$ and a reordering $\mathcal{R}$, we define the function $P : [r] \to [r]$ as follows:

$$P(i_1) = i_2 \iff \forall n \in [K+1] \ \forall j \in \mathbb{A}_n \ w^{(0)}_{i_1 j} = w^{(0)}_{i_2 \pi_n[j]} \text{ and } b^{(0)}_{i_1} = b^{(0)}_{i_2} \tag{19}$$

By our assumption on the initialization, for every $i_1 \in [r]$ there exists a unique $i_2 \in [r]$ such that $P(i_1) = i_2$. Therefore, $P$ is a well defined inverse function.

We will prove the following claim by induction on $t \ge 0$:

1. $\forall i \in [r] \ \left( \forall n \in [K+1] \ \forall j \in \mathbb{A}_n \ w^{(t)}_{ij} = w^{(t)}_{P(i)\pi_n[j]} \right) \text{ and } \left( b^{(t)}_i = b^{(t)}_{P(i)} \right)$

2. $\forall \boldsymbol{x} \in \mathbb{S}_x \ N(\boldsymbol{x}; \boldsymbol{W}^{(t)}, \boldsymbol{b}^{(t)}) = N(\boldsymbol{x}^\pi; \boldsymbol{W}^{(t)}, \boldsymbol{b}^{(t)})$

For $t = 0$, the first property is correct by the definition of $P$. By our initialization assumption, every $i \in [r]$ satisfies $\boldsymbol{w}^{(0)}_i \cdot \boldsymbol{x} = \boldsymbol{w}^{(0)}_{P(i)} \cdot \boldsymbol{x}^\pi$ and $b^{(0)}_i = b^{(0)}_{P(i)} = 0$. Then the second property can be proven by:

$$N(\boldsymbol{x}; \boldsymbol{W}^{(0)}, \boldsymbol{b}^{(0)}) = \sum_{i\in[r]} \sigma(\boldsymbol{w}^{(0)}_i \cdot \boldsymbol{x} + b^{(0)}_i) - 1 = \sum_{i\in[r]} \sigma(\boldsymbol{w}^{(0)}_{P(i)} \cdot \boldsymbol{x}^\pi + b^{(0)}_{P(i)}) - 1 = N(\boldsymbol{x}^\pi; \boldsymbol{W}^{(0)}, \boldsymbol{b}^{(0)}) \tag{20}$$

Assuming the correctness of the claim for $t-1$, we prove the claim for $t$. According to Eq. (5), for proving the first property, it is enough to show for every $i \in [r]$ that the following holds:

$$\forall n \in [K+1] \ \forall j \in \mathbb{A}_n \ \left[ w^{(t-1)}_i + \frac{\eta}{|\mathbb{S}|} \sum_{(\boldsymbol{x},y)\in\mathbb{G}^{(t-1)}_i} y\boldsymbol{x} \right]_j = \left[ w^{(t-1)}_{P(i)} + \frac{\eta}{|\mathbb{S}|} \sum_{(\boldsymbol{x},y)\in\mathbb{G}^{(t-1)}_{P(i)}} y\boldsymbol{x} \right]_{\pi_n[j]}$$

$$\left[ b^{(t-1)}_i + \frac{\eta}{|\mathbb{S}|} \sum_{(\boldsymbol{x},y)\in\mathbb{G}^{(t-1)}_i} y \right]_j = \left[ b^{(t-1)}_{P(i)} + \frac{\eta}{|\mathbb{S}|} \sum_{(\boldsymbol{x},y)\in\mathbb{G}^{(t-1)}_{P(i)}} y \right]_{\pi_n[j]} \tag{21}$$

By the induction assumption, $w^{(t-1)}_{ij} = w^{(t-1)}_{P(i)\pi_n[j]}$ and $b^{(t-1)}_{P(i)j} = b^{(t-1)}_{P(i)\pi_n[j]}$. Therefore, we can see that every $\boldsymbol{x} \in \mathbb{S}_x$ satisfies:

$$\boldsymbol{w}^{(t-1)}_i \cdot \boldsymbol{x} + b^{(t-1)}_i = \sum_{j\in[D]} w^{(t-1)}_{ij} x_j + b^{(t-1)}_i = \sum_{n\in[K+1]} \sum_{j\in\mathbb{A}_n} w^{(t-1)}_{ij} x_j + b^{(t-1)}_i \tag{22}$$

$$= \sum_{n\in[K+1]} \sum_{j\in\mathbb{A}_n} w^{(t-1)}_{P(i)\pi_n[j]} x^\pi_{\pi_n[j]} + b^{(t-1)}_{P(i)} = \sum_{j\in[D]} w^{(t-1)}_{P(i)j} x^\pi_j + b^{(t-1)}_{P(i)}$$

$$= \boldsymbol{w}^{(t-1)}_{P(i)} \cdot \boldsymbol{x}^\pi + b^{(t-1)}_{P(i)}$$

Recall, $y = y^\pi$, then by the induction assumption and Definition A.1 the following holds:

$$x \in \mathbb{G}_i^{(t-1)} \Longleftrightarrow 1 - yN(x; W^{(t-1)}, b^{(t-1)}) > 0 \ \wedge \ w_i^{(t-1)} \cdot x + b_i^{(t-1)} > 0 \tag{23}$$

$$\Longleftrightarrow 1 - y^\pi N(x^\pi; W^{(t-1)}, b^{(t-1)}) > 0 \ \wedge \ w_{P(i)}^{(t-1)} \cdot x^\pi + b_{P(i)}^{(t-1)} > 0 \Longleftrightarrow x^\pi \in \mathbb{G}_{P(i)}^{(t-1)}$$

Therefore,

$$\left[ \sum_{(x,y) \in \mathbb{G}_i^{(t-1)}} yx \right]_j = \left[ \sum_{(x,y) \in \mathbb{G}_{P(i)}^{(t-1)}} yx \right]_{\pi_n[j]} \quad \text{and} \quad \left[ \sum_{(x,y) \in \mathbb{G}_i^{t-1}} y \right]_j = \left[ \sum_{(x,y) \in \mathbb{G}_{P(i)}^{t-1}} y \right]_{\pi_n[j]} \tag{24}$$

We conclude that Eq. (21) is correct, and as a result the first claim in our proof by induction is correct. Therefore, every pair $x, x^\pi \in \mathbb{S}_x$ satisfies $w_i^{(t)} \cdot x = w_{P(i)}^{(t)} \cdot x^\pi$, by the first claim and Definition F.2. Similarly, it holds that $b_i^{(t)} = b_{P(i)}^{(t)}$. Therefore:

$$N(x; W^{(t)}, b^{(t)}) = \sum_{i \in [r]} \sigma(w_i^{(t)} \cdot x + b_i^{(t)}) - 1 = \sum_{i \in [r]} \sigma(w_{P(i)}^{(t)} \cdot x^\pi + b_{P(i)}^{(t)}) - 1 = N(x^\pi; W^{(t)}, b^{(t)}) \tag{25}$$

which completes the proof.

### F.2 PROOF OF THEOREM 7.1

We will prove the claim by induction on $t$. For $t = T$ the claim is correct by the claim's assumption. Assuming the correctness of the claim for $t - 1$, we will prove it for $t$. By Eq. (5):

$$w_i^{(t)} = w_i^{(t-1)} + \frac{\eta}{|\mathbb{S}_x|} \sum_{(x,y) \in \mathbb{G}_{i_1}^{(t-1)}} yx \tag{26}$$

Given $j_1, j_2 \in \mathbb{A}_n$, by the induction assumption $w_{ij_1}^{(t-1)} = w_{ij_2}^{(t-1)}$. Then, $w_{ij_1}^{(t)} = w_{ij_2}^{(t)}$ if and only if

$$\left[ \sum_{(x,y) \in \mathbb{G}_i^{(t-1)}} yx \right]_{j_1} = \left[ \sum_{(x,y) \in \mathbb{G}_i^{(t-1)}} yx \right]_{j_2} \tag{27}$$

We define the following ordering $\mathcal{R}$:

1. $\forall n' \in [K+1] \backslash \{n\} \ \forall j \in \mathbb{A}_n \ \ \pi_{n'}[j] = j$

2. $\forall j \in \mathbb{A}_n \backslash \{j_1, j_2\} \ \pi_n[j] = j \ \textbf{and} \ \pi_n[j_1] = j_2 \ \pi_n[j_2] = j_1$

In other words, $\pi$ is the permutation that switches $j_1$ and $j_2$. Since $w_{ij_1}^{(t-1)} = w_{ij_2}^{(t-1)}$, we can determine that $w_i \cdot x = w_i \cdot x^\pi$. Recall that $N(x; W^{(t)}, b^{(t)}) = N(x^\pi; W^{(t)}, b^{(t)})$ by Lemma F.1. Thus, $(x, y) \in \mathbb{G}_i^{(t-1)} \Longleftrightarrow (x^\pi, y^\pi) \in \mathbb{G}_i^{(t-1)}$ by Definition A.1. Then, using the fact that $\forall (x, y) \in \mathbb{G}_i^{(t-1)} \ yx_{j_1} = y^\pi x_{\pi_n[j_1]}^\pi = yx_{j_2}^\pi$ we can conclude that:

$$\left[ \sum_{(x,y) \in \mathbb{G}_i^{(t-1)}} yx \right]_{j_1} = \left[ \sum_{(x,y) \in \mathbb{G}_i^{(t-1)}} yx \right]_{j_2}$$

as required. □

## G PROOF OF THEOREM 7.2 (MINIMUM $l_2$-NORM IS A DNF-RECOVERY SOLUTION)

We set out to characterize the structure of the min-norm solution. In the proof we use Definition D.1 where $\mathbb{A}_{K+1} = [D] \backslash \cup_{n \in [K]} \mathbb{A}_n$. Following the notation from Definition B.1, we say that a solution

$(\boldsymbol{W}, \boldsymbol{b})$ satisfies the $MIN_+$ property for a positive point $\boldsymbol{x} \in \mathbb{S}_p$ if there exists $\mathbb{I} \subseteq [r]$ such that $\sum_{i \in \mathbb{I}} \boldsymbol{w}_i \cdot \boldsymbol{x} + b_i \geq 2$. We define $\mathbb{U}$ to be the set of global minima of the loss in Eq. (2). We first prove several lemmas.

## G.1 AUXILIARY LEMMAS

**Definition G.1.** *For a term $n \in [K]$, we define the special sample $\boldsymbol{x}^{(n)} \in \mathbb{S}_x$ of this term as:*

$$\forall j \in \mathbb{A}_n \ x_j^{(n)} = 1 \text{ and } \forall j \in [D] \backslash \mathbb{A}_n \ x_j^{(n)} = -1 \tag{28}$$

*We denote the set of all the special samples by $\mathbb{O} = \{\boldsymbol{x} \in \mathbb{S}_p \mid \exists n \in [K] \ \boldsymbol{x} = \boldsymbol{x}^{(n)}\}$*

**Lemma G.1.** *Given $(\boldsymbol{W}, \boldsymbol{b})$, assume the following conditions are satisfied:*

1. *$\forall i \in [r], \ \forall j \in [D] \ \boldsymbol{w}_{ij} \geq 0$.*

2. *Every $\boldsymbol{x} \in \mathbb{O}$ satisfies the $MIN_+$ property.*

*Then $(\boldsymbol{W}, \boldsymbol{b})$ satisfies the $MIN_+$ property for all positive samples.*

*Proof.* Let $\boldsymbol{x} \in \mathbb{S}_p$. Then $\exists n \in [K]$ such that $\forall j \in \mathbb{A}_n \ \boldsymbol{x}_j = 1$. According to the second assumption, $\exists \mathbb{I} \subseteq [r]$ such that $\sum_{i \in \mathbb{I}} \boldsymbol{w}_i \cdot \boldsymbol{x}^{(n)} + b_i \geq 2$.

For every $i \in [r]$ the following holds:

$$\boldsymbol{w}_i \cdot \boldsymbol{x} = \sum_{j \in [D]} w_{ij} x_j = \sum_{j \in \mathbb{A}_j} w_{ij} + \sum_{j \in [D] \backslash \mathbb{A}_n} x_j w_{ij} \tag{29}$$

From the first condition of the claim we can deduce that

$$\sum_{j \in \mathbb{A}_j} w_{ij} + \sum_{j \in [D] \backslash \mathbb{A}_n} x_j w_{ij} \geq \sum_{j \in \mathbb{A}_n} w_{ij} - \sum_{j \in [D] \backslash \mathbb{A}_j} w_{ij} = \sum_{j \in [D]} w_{ij} x_j^{(n)} = \boldsymbol{w}_i \cdot \boldsymbol{x}^{(n)} \tag{30}$$

Then:

$$\sum_{i \in \mathbb{I}} \sigma(\boldsymbol{w}_i \cdot \boldsymbol{x} + b_i) \geq \sum_{i \in \mathbb{I}} \sigma(\boldsymbol{w}_i \cdot \boldsymbol{x}^{(n)} + b_i) \geq 2 \tag{31}$$

and $(\boldsymbol{W}, \boldsymbol{b})$ satisfies the $MIN_+$ property for $\boldsymbol{x}$ as required. $\qquad \square$

The following definition will be very useful in our analysis.

**Definition G.2.** *Given a min-norm solution $(\boldsymbol{W}^*, \boldsymbol{b}^*)$, then we say that the solution $(\hat{\boldsymbol{W}}, \hat{\boldsymbol{b}})$ is an $i$-modified solution if the following holds:*

$$\forall i' \in [r] \backslash \{i\} \ \hat{\boldsymbol{w}}_{i'} = \boldsymbol{w}_{i'}^* \text{ and } \hat{b}_{i'} = b_{i'}^* \tag{32}$$

Thus, given a min-norm solution, to define an $i$-modified solution, we only need to define the neuron $(\boldsymbol{w}_i, b_i)$.

**Lemma G.2.** *Given $(\boldsymbol{W}^*, \boldsymbol{b}^*)$, then every $i \in [r]$ satisfies:*

1. *$b_i = BT(\boldsymbol{w}_i)$.*

2. *$\forall j \in [D] \ \boldsymbol{w}_{ij} \geq 0$.*

3. *$\exists n \in [K]$ such that $\forall j \in \mathbb{A}_n \ \boldsymbol{w}_{ij} \geq 0$ and $\forall j \in [D] \backslash \mathbb{A}_n \ \boldsymbol{w}_{ij} = 0$.*

*Proof.* Given $(\boldsymbol{W}^*, \boldsymbol{b}^*)$ and $i \in [r]$, we will prove the properties one by one.

**Property 1 -** Assume by contradiction that $b_i^* \neq BT(\boldsymbol{w}_i^*)$. By Lemma B.2, $b_i$ has to be smaller than $BT(\boldsymbol{w}_i^*)$, because otherwise $(\boldsymbol{W}^*, \boldsymbol{b}^*)$ is not a global minimum of the loss in Eq. (2). Now consider the $i$-modified solution, $(\hat{\boldsymbol{W}}, \hat{\boldsymbol{b}})$, which is defined by:

$$\hat{\boldsymbol{w}}_i = \boldsymbol{w}_i^*, \quad \hat{b}_i = BT(\hat{\boldsymbol{w}}) \tag{33}$$

By the assumption $\hat{b}_i > b_i^*$. Then, every $\boldsymbol{x} \in \mathbb{S}_x$ satisfies the following:

$$\boldsymbol{x} \cdot \boldsymbol{w}_i^* + b_i < \boldsymbol{x} \cdot \hat{\boldsymbol{w}}_i + \hat{b}_i \tag{34}$$

Because $(\boldsymbol{W}^*, \boldsymbol{b}^*)$ satisfies the MIN$_+$ property, then $(\hat{\boldsymbol{W}}, \hat{b})$ satisfies it as well. Recall that $(\boldsymbol{W}^*, \boldsymbol{b}^*)$ also satisfies the MIN$_-$ property, thus from Definition G.2 every $\boldsymbol{x} \in \mathbb{S}_n$ satisfies:

$$\forall i' \in [r]\backslash\{i\} \quad 0 > \boldsymbol{x} \cdot \boldsymbol{w}_{i'}^* + b_{i'} = \boldsymbol{x} \cdot \hat{\boldsymbol{w}}_{i'} + \hat{b}_{i'} \tag{35}$$

In addition, from Lemma D.1 we know that $0 > \boldsymbol{x} \cdot \hat{\boldsymbol{w}}_i + \hat{b}_i$. Therefore, $(\hat{\boldsymbol{W}}, \hat{b})$ satisfies the MIN$_-$ property. By Lemma B.1, $(\hat{\boldsymbol{W}}, \hat{b}) \in \mathbb{U}$. From Definition B.4 the bias threshold is a negative, then $b_i^* < \hat{b}_i \leq 0 \rightarrow |\hat{b}_i| < |b_i^*|$. We know that $\hat{\boldsymbol{w}}_i = \boldsymbol{w}_i^*$ the $\|(\hat{\boldsymbol{w}}_i, \hat{b}_i)\|_2^2 < \|(\boldsymbol{w}_i^*, b_i^*)\|_2^2$ which is in contradiction to the optimality of $(\boldsymbol{W}^*, \boldsymbol{b}^*)$.

**Property 2 -** Assume by contradiction that $\exists j' \in [D]$ such that $w_{ij'}^* < 0$. Consider the following $i$-modified solution $(\hat{\boldsymbol{W}}, \hat{b})$:

$$\forall j \in [D]\backslash\{j'\} \ \hat{\boldsymbol{w}}_{ij} = \boldsymbol{w}_{ij}^* \ \wedge \ \hat{\boldsymbol{w}}_{ij'} = 0 \ \wedge \ \hat{b}_i = BT(\hat{\boldsymbol{w}}_i) \tag{36}$$

We want to show that:

$$\sum_{n \in [K]} V_n(\boldsymbol{w}_i^*) = \sum_{n \in [K]} V_n(\hat{\boldsymbol{w}}_i) \tag{37}$$

If $\exists n' \in [K]$ such that $j' \in \mathbb{A}_{n'}$, then it follows that $V_{n'}(\boldsymbol{w}_i^*) = V_{n'}(\hat{\boldsymbol{w}}_i) = 0$ and Eq. (37) is satisfied. Otherwise, $j' \in \mathbb{A}_{K+1}$, by Definition D.1, the indices of $\mathbb{A}_{K+1}$ don't affect the value of the sums in Eq. (37). Therefore, this equation is satisfied in this case as well.

We know that $b_i^* = BT(\boldsymbol{w}_i^*)$ from the first property, thus every $\boldsymbol{x} \in \mathbb{S}_x$ satisfies the following:

$$\boldsymbol{x} \cdot \boldsymbol{w}_i^* + b_i^* = \sum_{j \in [D]} x_j w_{ij}^* + BT(\boldsymbol{w}_i^*) \tag{38}$$

$$= \sum_{j \in [D]\backslash\{j'\}} x_j w_{ij}^* + x_{j'} w_{ij'}^* - |w_{ij'}^*| - \sum_{j \in [D]\backslash\{j'\}} |w_{ij}^*| + \sum_{n \in [K]} 2V_n(\boldsymbol{w}_i^*)$$

We can see that $x_{j'} w_{ij'}^* - |w_{ij'}^*| \leq 0$ for every $x_{j'}$. Then,

$$\sum_{j \in [D]\backslash\{j'\}} x_j w_{ij}^* + x_{j'} w_{ij'}^* - |w_{ij'}^*| - \sum_{j \in [D]\backslash\{j'\}} |w_{ij}^*| + \sum_{n \in [K]} 2V_n(\boldsymbol{w}_i^*)$$

$$\leq \sum_{j \in [D]\backslash\{j'\}} x_j w_{ij}^* - \sum_{j \in [D]\backslash\{j'\}} |w_{ij}^*| + \sum_{n \in [K]} 2V_n(\boldsymbol{w}_i^*) \tag{39}$$

$$= \sum_{j \in [D]} x_j \hat{w}_{ij} - \|\hat{\boldsymbol{w}}_i\|_1 + \sum_{n \in [K]} 2V_n(\hat{\boldsymbol{w}}_i) = \boldsymbol{x} \cdot \hat{\boldsymbol{w}}_i + BT(\hat{\boldsymbol{w}}_i) = \boldsymbol{x} \cdot \hat{\boldsymbol{w}}_i + \hat{b}_i$$

Using the fact that $\boldsymbol{x} \cdot \boldsymbol{w}_i^* + b_i^* \leq \boldsymbol{x} \cdot \hat{\boldsymbol{w}}_i + \hat{b}_i$ with the fact that $(\boldsymbol{W}^*, \boldsymbol{b}^*)$ satisfies the MIN$_+$ property, we can conclude that $(\hat{\boldsymbol{W}}, \hat{b})$ satisfies this property too.

According to the "Property 1" above and Definition G.2 we have:

$$\forall i' \in [r]\backslash\{i\} \ BT(\hat{\boldsymbol{w}}_{i'}) = BT(\boldsymbol{w}_{i'}^*) = b_{i'}^* = \hat{b}_{i'} \tag{40}$$

In addition, we know that $\hat{b}_i = BT(\hat{\boldsymbol{w}}_i)$ by Eq. (36). According to Lemma D.1, the solution satisfies the MIN$_-$ property and from Lemma B.1, $(\hat{\boldsymbol{W}}, \hat{b}) \in \mathbb{U}$

From Eq. (36), we know that $|w_{ij'}^*| > |\hat{w}_{ij'}| \rightarrow \|\boldsymbol{w}_i^*\|_1 > \|\hat{\boldsymbol{w}}_i\|_1$. Combining this with Eq. (37) we can conclude that $|BT(\boldsymbol{w}_i^*)| > |BT(\hat{\boldsymbol{w}}_i)| \rightarrow |b_i^*| > |\hat{b}_i|$. Therefore, $\|(\hat{\boldsymbol{w}}_i, \hat{b}_i)\|_2^2 < \|(\boldsymbol{w}_i^*, b_i^*)\|_2^2$ in contradiction to the optimality of $(\boldsymbol{W}^*, \boldsymbol{b}^*)$.

**Property 3 -** Assume by contradiction that there exists $i \in [r]$ such that:

$$\exists n_1, n_2 \in [K+1] \text{ such that } \exists j \in \mathbb{A}_{n_1} \ w_{ij} > 0 \text{ and } \exists j \in \mathbb{A}_{n_2} \ w_{ij} > 0 \tag{41}$$

Without loss of generality we assume:

$$\sum_{j \in \mathbb{A}_{n_1}} w_{ij} \geq \sum_{j \in \mathbb{A}_{n_2}} w_{ij} \tag{42}$$

Let's look on the following $i$-modified $(\hat{\boldsymbol{W}}, \hat{\boldsymbol{b}})$ which is defined by:

$$\forall j \in \mathbb{A}_{n_2} \ \hat{w}_{ij} = 0 \text{ and } \forall j \in [D] \backslash \mathbb{A}_{n_2} \ \hat{w}_{ij} = w_{ij} \text{ and } \hat{b}_i = BT(\hat{\boldsymbol{w}}_i) \tag{43}$$

First, we will show that $\hat{b}_i \geq b_i^*$. If $n_2 \neq K+1$, from Definition B.4 and the assumption that $|\mathbb{A}_{n_2}| > 1$, the following holds:

$$\hat{b}_i = BT(\hat{\boldsymbol{w}}_i) = BT(\boldsymbol{w}_i^*) + \sum_{j \in \mathbb{A}_{n_2}} |w_{ij}| - 2V_{n_2}(\boldsymbol{w}_i^*) = b_i + \sum_{j \in \mathbb{A}_{n_2}} |w_{ij}^*| - 2V_{n_2}(\boldsymbol{w}_i) \geq b_i^* \tag{44}$$

Otherwise $n_2 = K + 1$ and from Definition B.4 the following holds:

$$\hat{b}_i = BT(\hat{\boldsymbol{w}}_i) = BT(\boldsymbol{w}_i^*) + \sum_{j \in \mathbb{A}_{n_2}} |w_{ij}| = b_i + \sum_{j \in \mathbb{A}_{n_2}} |w_{ij}^*| \geq b_i^* \tag{45}$$

In both cases $\hat{b}_i \geq b_i^*$ as required.

Given $\widetilde{n} \in [K]$, we know that $(\boldsymbol{W}^*, \boldsymbol{b}^*) \in \mathbb{U}$ and thus it satisfies the MIN$_+$ property for $\boldsymbol{x}^{(\widetilde{n})}$. Then, $\exists I \subseteq [r]$ such that:

$$\sum_{i' \in I} \boldsymbol{w}_{i'}^* \cdot \boldsymbol{x} + b_{i'}^* \geq 2 \tag{46}$$

We will show that $(\hat{\boldsymbol{W}}, \hat{\boldsymbol{b}})$ satisfies this property for $\widetilde{n}$ as well. If $\widetilde{n} \neq n_2$, due to the first properly of this lemma and the fact that $\hat{b}_i \geq b_i^*$ the following holds:

$$\boldsymbol{w}_i^* \cdot \boldsymbol{x}^{(\widetilde{n})} + b_i^* = \sum_{j \in [D] \backslash \mathbb{A}_{n_2}} w_{ij}^* x_j^{(\widetilde{n})} - \sum_{j \in \mathbb{A}_{n_2}} w_i^* + b_i^* < \sum_{j \in [D] \backslash \mathbb{A}_{n_2}} w_{ij}^* x_j^{(\widetilde{n})} + b_i^* \leq \sum_{j \in [D]} \hat{w}_{ij} x_j^{(\widetilde{n})} + \hat{b}_i$$
$$= \hat{\boldsymbol{w}}_i \cdot \boldsymbol{x}^{(\widetilde{n})} + \hat{b}_i \tag{47}$$

Then we can conclude that $(\hat{\boldsymbol{W}}, \hat{\boldsymbol{b}})$ satisfies the MIN$_+$property for $\boldsymbol{x}^{(\widetilde{n})}$ from the following inequalities:

$$\sum_{i' \in \mathbb{I}} \hat{\boldsymbol{w}}_{i'} \cdot \boldsymbol{x} + \hat{b}_{i'} \geq \sum_{i' \in \mathbb{I}} \boldsymbol{w}_{i'}^* \cdot \boldsymbol{x} + b_{i'}^* \geq 2 \tag{48}$$

Otherwise, $\widetilde{n} = n_2$. By the fact that $b_i^* = BT(\boldsymbol{w}_i^*) \leq 0$, the first property of this lemma and Eq. (42) the following holds:

$$\boldsymbol{w}_i^* \cdot \boldsymbol{x} + b_i^* = \sum_{j \in \mathbb{A}_{n_2}} w_{ij}^* - \sum_{j \in [D] \backslash \mathbb{A}_{n_2}} w_{ij}^* + b_i^* \leq \sum_{j \in \mathbb{A}_{n_2}} w_{ij}^* - \sum_{j \in \mathbb{A}_{n_1}} w_{ij}^* + b_i^* \leq \sum_{j \in \mathbb{A}_{n_2}} w_{ij}^* - \sum_{j \in \mathbb{A}_{n_1}} w_{ij}^* < 0 \tag{49}$$

Therefore, using Definition G.2, $(\hat{\boldsymbol{W}}, \hat{\boldsymbol{b}})$ satisfies the following:

$$\sum_{i' \in \mathbb{I} \backslash \{i\}} \hat{\boldsymbol{w}}_{i'} \cdot \boldsymbol{x} + \hat{b}_{i'} = \sum_{i' \in \mathbb{I} \backslash \{i\}} \boldsymbol{w}_{i'}^* \cdot \boldsymbol{x} + b_{i'}^* \geq \sum_{i' \in \mathbb{I}} \boldsymbol{w}_{i'}^* \cdot \boldsymbol{x} + b_{i'}^* \geq 2 \tag{50}$$

We can conclude that $(\hat{\boldsymbol{W}}, \hat{\boldsymbol{b}})$ satisfies the MIN$_+$ property for $\boldsymbol{x}^{(\widetilde{n})}$. Combining this with the first property we can see that $(\hat{\boldsymbol{W}}, \hat{\boldsymbol{b}})$ meets the condition of Lemma G.1 and then it satisfies the MIN$_+$ property for all positive samples.

According to the first property and Definition G.2:

$$\forall i' \in [r] \backslash \{i\} \ BT(\hat{\boldsymbol{w}}_{i'}) = BT(\boldsymbol{w}_{i'}^*) = b_{i'}^* = \hat{b}_{i'} \tag{51}$$

In addition, we know that $\hat{b}_i = BT(\hat{\boldsymbol{w}}_i)$ by Eq. (43). According to Lemma D.1, the solution satisfies the MIN$_-$ property and from Lemma B.1, $(\hat{\boldsymbol{W}}, \hat{b}) \in \mathbb{U}$.

As we saw $0 \geq b_i^* > \hat{b}_i \to |\hat{b}_i| > |b_i^*|$ and $\forall j \in \mathbb{A}_{n_2} \ w_{ij}^* \geq 0 = \hat{w}_{ij}$ then $\|(\boldsymbol{w}_i^*, b_i^*)\|_2^2 > \|(\hat{\boldsymbol{w}}_i, \hat{b}_i)\|_2^2$ in contradiction to the optimality of $(\boldsymbol{W}^*, \boldsymbol{b}^*)$. $\qquad \square$

We next prove the following covering lemma.

**Lemma G.3.** *Given* $(\boldsymbol{W}^*, \boldsymbol{b}^*)$*, every* $i \in [r]$ *covers some term* $n \in [K]$ *or* $\boldsymbol{w}_i = 0$

*Proof.* Given $i \in [r]$, if $\boldsymbol{w}_i = 0$ the claim is true. Otherwise, by the third property of Lemma G.2, $\exists n \in [r]$ such that:

$$\forall j \in \mathbb{A}_n \ \ \boldsymbol{w}_{ij} > 0 \text{ and } \forall j \in [D]\backslash\mathbb{A}_n \ \ \boldsymbol{w}_{ij} = 0 \tag{52}$$

We will prove the claim for $n$ by assuming by contradiction that:

$$\exists j_1, j_2 \in \mathbb{A}_n \text{ such that } w_{ij_1}^* \neq w_{ij_2}^* \tag{53}$$

Without loss of generality, we assume $w_{ij_1}^* > w_{ij_2}^*$ and $w_{ij_2}^* = \min_{j \in \mathbb{A}_n}\{w_{ij}^*\}$.

Define the following $i$-modified solution $(\hat{\boldsymbol{W}}, \hat{\boldsymbol{b}})$:

$$\forall j \in \mathbb{A}_n \ \hat{w}_{ij} = w_{ij_2}^* \text{ and } \forall j \in [D]\backslash\mathbb{A}_n \ \hat{w}_{ij} = w_{ij}^* \text{ and } \hat{b}_i = BT(\hat{\boldsymbol{w}}_i) \tag{54}$$

Note that $V_n(\boldsymbol{w}_i^*) = V_n(\hat{\boldsymbol{w}}_i) = w_{ij_2}^*$.

Given $\boldsymbol{x} \in \mathbb{S}_p$, we know that $(\boldsymbol{W}^*, \boldsymbol{b}^*)$ satisfies the MIN$_+$ property for $\boldsymbol{x}$. Then, $\exists I \subseteq [r]$ such that:

$$\sum_{i' \in I} \boldsymbol{w}_{i'}^* \cdot \boldsymbol{x} + b_{i'}^* \geq 2 \tag{55}$$

If $\boldsymbol{x} \cdot \boldsymbol{t}_n^* = \|\boldsymbol{t}_n^*\|_1$, due to the third property of Lemma G.2 the following holds:

$$\boldsymbol{w}_i^* \cdot \boldsymbol{x} + b_i^* = \sum_{j \in \mathbb{A}_n} w_{ij}^* + BT(\boldsymbol{w}_i^*) = \sum_{j \in \mathbb{A}_n} w_{ij}^* - \sum_{j \in \mathbb{A}_n} |w_{ij}^*| + 2V_n(\boldsymbol{w}_i^*) = 2w_{ij_2}^* \tag{56}$$

$$= \sum_{j \in \mathbb{A}_n} w_{ij}^* - \sum_{j \in \mathbb{A}_n} w_{ij}^* + 2w_{ij_2}^* = \sum_{j \in \mathbb{A}_n} \hat{w}_{ij} - \|\hat{\boldsymbol{w}}\|_1 + 2V_n(\hat{\boldsymbol{w}}_i) = \hat{\boldsymbol{w}}_i \cdot \boldsymbol{x} + \hat{b}_i$$

Then we can conclude that $(\hat{\boldsymbol{W}}, \hat{\boldsymbol{b}})$ satisfies this property for $\boldsymbol{x}$ from the following:

$$\sum_{i' \in \mathbb{I}} \hat{\boldsymbol{w}}_{i'} \cdot \boldsymbol{x} + \hat{b}_{i'} = \sum_{i' \in \mathbb{I}} \boldsymbol{w}_{i'}^* \cdot \boldsymbol{x} + b_{i'}^* \geq 2 \tag{57}$$

Otherwise, $\boldsymbol{x} \cdot \boldsymbol{t}_n^* < \|\boldsymbol{t}_n^*\|_1$ and by Definition B.4:

$$\boldsymbol{w}_i^* \cdot \boldsymbol{x} + b_i = \sum_{j \in \mathbb{A}_n} w_{ij}^* x_j + BT(\boldsymbol{w}_i^*) \leq 0 \tag{58}$$

Therefore, using Definition G.2, we have that $(\hat{\boldsymbol{W}}, \hat{\boldsymbol{b}})$ satisfies:

$$\sum_{i' \in \mathbb{I}\backslash\{i\}} \hat{\boldsymbol{w}}_{i'} \cdot \boldsymbol{x} + \hat{b}_{i'} = \sum_{i' \in \mathbb{I}\backslash\{i\}} \boldsymbol{w}_{i'}^* \cdot \boldsymbol{x} + b_{i'}^* \geq \sum_{i' \in \mathbb{I}} \boldsymbol{w}_{i'}^* \cdot \boldsymbol{x} + b_{i'}^* \geq 2 \tag{59}$$

We can conclude that $(\hat{\boldsymbol{W}}, \hat{\boldsymbol{b}})$ satisfies the MIN$_+$ property for $\boldsymbol{x}$.

According to the first property of Lemma G.2 and Definition G.2:

$$\forall i' \in [r]\backslash\{i\} \ \ BT(\hat{\boldsymbol{w}}_{i'}) = BT(\boldsymbol{w}_{i'}^*) = b_{i'}^* = \hat{b}_{i'} \tag{60}$$

In addition, we know that $\hat{b}_i = BT(\hat{\boldsymbol{w}}_i)$ by Eq. (54). According to Lemma D.1, the solution satisfies the MIN$_-$ property and from Lemma B.1, $(\hat{\boldsymbol{W}}, \hat{b}) \in \mathbb{U}$

Finally, we can see that, $\|\boldsymbol{w}_i^*\|_1 > \|\hat{\boldsymbol{w}}_i\|_1$ implies that $|BT(\boldsymbol{w}_i)| > |BT(\hat{\boldsymbol{w}}_i)|$ and $\forall j \in [D] w_{ij}^* \geq \hat{w}_{ij}$. Thus, we have $\|(\boldsymbol{w}_i^*, b_i^*)\|_2^2 > \|(\hat{\boldsymbol{w}}_i, \hat{b}_i)\|_2^2$. This is contradiction to the optimality of $(\boldsymbol{W}^*, \boldsymbol{b}^*)$, as desired.

We can now define $\lambda_i = V_n(\boldsymbol{w}_i^*)$ and we know that the neuron $i$ satisfies:

$$\forall j \in \mathbb{A}_n \ \ w_{ij}^* = \lambda_i \text{ and } \forall j \in [D]\backslash\mathbb{A}_n \ \ w_{ij}^* = 0 \tag{61}$$

Therefore, $\boldsymbol{w}_i = \lambda_i \boldsymbol{t}_n^*$ and we can say that the neuron $i$ covers the term $n$. $\qquad \square$

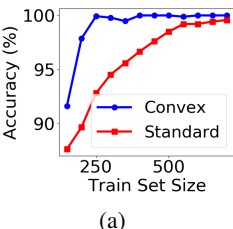 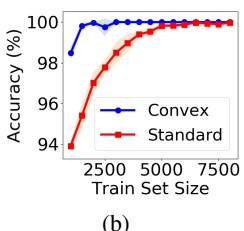 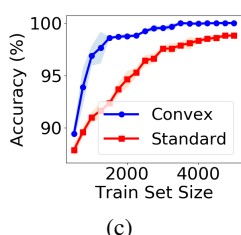

|       |       |       |
| :---: | :---: | :---: |
|  (a)  |  (b)  |  (c)  |

Figure 4: Test accuracy for the convex and standard networks. (a) $D = 10$ and the target DNF has 3 terms of size: (2,3,3) (b) $D = 27$ and the target DNF has 3 terms of size 3, 2 terms of size 2 and one term of size 6. (c) $D = 35$ and the target DNF has 3 terms of size 2, 4 terms of size 4 and 2 terms of size 2.

## G.2  FINISHING THE PROOF OF THEOREM 7.2

*Proof.* We know from Lemma G.3 that each neuron covers some term. We denote the term that is covered by neuron $i$ as $n_i$. Given $n \in [K]$, we assume by contradiction that it is not covered, namely $\forall i \in [r]\ n_i \neq n$. Consider the special positive sample $\boldsymbol{x}^{(n)}$ for which it holds that:

$$\forall i \in [r]\quad \boldsymbol{x}^{(n)} \cdot \boldsymbol{w}_i^* = \lambda_n \boldsymbol{x}^{(n)} \cdot \boldsymbol{t}_{n_i}^* = -\lambda_n \|\boldsymbol{t}_{n_i}^*\|_1 < 0 \tag{62}$$

By the first property of Lemma G.3, $\forall i \in [r]$, $b_i^* = BT(\boldsymbol{w}_i^*) \leq 0$. Therefore, $\forall i \in [r]\ \boldsymbol{x}^{(n)} \cdot \boldsymbol{w}_i^* + b_i^* < 0$ in contradiction to the fact that $(\boldsymbol{W}^*, \boldsymbol{b}^*) \in \mathbb{U}$

By Definition 5.2, $\boldsymbol{W}$ covers the DNF $f^*$. Together with Lemma G.3, this implies that $\boldsymbol{W}^*$ is a DNF-recovery solution. $\square$

## H  EXPERIMENT DETAILS AND ADDITIONAL RESULTS

In the experiments for Figure 3a and Figure 3b, we use a small Gaussian initialization $\boldsymbol{W}^{(0)} \sim \mathcal{N}(0, 10^{-5})$ and $\boldsymbol{b}^{(0)} = [0]^D$. The learning rate for GD is $\eta = 10^{-2}$, and the number of hidden units is $r = 700$. We create the train set by sampling uniformly from $[\pm 1]^D$. The test set is size $10^4$, sampled uniformly from $[\pm 1]^D$.

In the experiments of Figure 3a, for every train set size we run the experiment five times and report average accuracy. If GD converges to a local minimum during learning we re-initialize and re-train. Therefore, all our results are taken when the train error is zero.

For the tabular datasets, we consider the three UCI datasets: kr-vs-kp, Splice, and diabetes. For all these, we convert the input into binary by changing categorical variables to one-hot. We also consider binary classification so in kr-vs-kp the class 'won' is positive considered and 'notwon' is negative, in Splice the classes 'EI' and 'IE' are considered positive and 'N' negative, and diabetes is binary by design. We train on $90\%$ of the data and test on $10\%$.

### H.1  ADDITIONAL EXPERIMENTS

Here we report additional experiments on various $D$ values and DNFs. Figure 4 reports results on test accuracy for convex and standard networks, for $D = 10, 27, 35$. We note that for $D > 40$ values the standard network converges very slowly and often does not converge to zero error, and thus we do not include comparisons in these cases.

Figure 5 reports results on DNF recovery accuracy for $D = 10, 30, 100$. For $D \geq 30$, the standard network typically fails to recover the DNF.

Figure 6 shows learning with small and large initial scale for an unbalanced DNF. To create the figures of the global minima, we use hierarchical clustering to cluster the neurons.

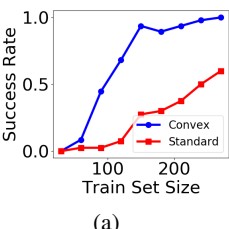 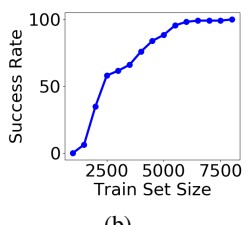 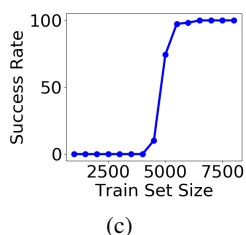

(a)                        (b)                        (c)

Figure 5: Reconstruction success rate for the convex and standard networks. (a) D = 10 and the target DNF has 3 terms of size: (2,3,3) (b) D=30 and the target DNF has 5 terms of size 6. Standard network fails to reconstruct (success rate below 5%). (c) D = 100 and the target DNF has 15 terms of size 5. Standard network fails to reconstruct (successs rate below 5%).

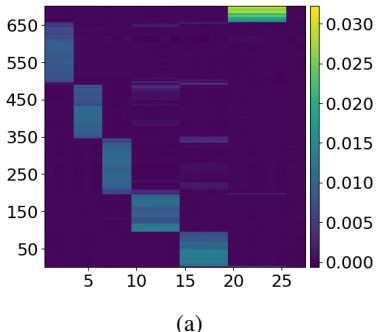 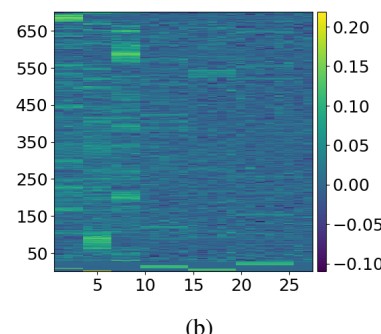

(a)                                 (b)

Figure 6: Model learned by a convex network for DNF with $D = 27$ for different initailization scales. The ground-trtuh DNF has 3 terms of size 3, 2 terms of size 2 and one term of size 1. (a) Small initialization - Can be seen to lead to good recovery. (b) Large initialization - Does not lead to good recovery.

## I  DETAILS FOR COMPUTER ASSISTED PROOF

The set $\mathbb{F}$ for which we proved the result consists of all balanced DNFs with input dimension $4 \leq D \leq 15$ with terms of sizes at least two, including ones where variables are not part of any term (e.g., when $D = 6$, we consider also the DNF: $(x_1 \wedge x_2) \vee (x_3 \wedge x_4)$).

We consider the following parameter ranges in the proof: $\gamma \in \{0.4, 0.5, 0.9\}$ and $\beta \in \{0.4, 0.5, 0.9\}$. For $4 \leq D \leq 12$ we used $\eta = 10^{-5}$ and $\epsilon \in \{10^{-6}, 2 \cdot 10^{-6}, 3 \cdot 10^{-6}, 4 \cdot 10^{-6}, 5 \cdot 10^{-6}, 6 \cdot 10^{-6}, 7 \cdot 10^{-6}, 8 \cdot 10^{-6}\}$. For $13 \leq D \leq 15$ we used $\eta = 10^{-4}$ and $\epsilon = 2 \cdot 10^{-6}$ to avoid long run time.[8]

Given $D, \eta, \epsilon$ and target function $f^*$ we use lemma $\alpha$-GD to learn $f^*$. This allows us to perform the simulation in integers without floating point inaccuracies. According to lemma 6.1 this is equivalent to learning with standard GD in term of convergence and recovery. We choose $\alpha = \frac{2^D}{\eta} \cdot 10^6$ such that the initialization is integer and by Eq. (5) the update step is in integer steps. We use int64 to avoid integer overflow problem.

The procedure begins by initializing $\boldsymbol{W} = [\pm \alpha \epsilon]^D$ and $\boldsymbol{b} = [0]^D$. Then we apply Eq. (5) using matrix multiplication in tensorflow on GPU, until we achieve 0 loss. After the network converges to a global minimum, we execute $\gamma$-pruning. To keep the simulation in integers we use the following trick. Instead of calculating if $\|\boldsymbol{w}_i\|_\infty$ is larger than $\gamma M_\infty(\boldsymbol{W})$, we calculate $10 \cdot \|\boldsymbol{w}_i\|_\infty$ and $10 \cdot \gamma M_\infty(\boldsymbol{W})$, because $10\gamma$ is an integer. Finally, we apply $\beta$-reconstruction to the pruned network to get the reconstructed network, using the same trick to do the simulation in integers.

---

[8]In the case of $D = 13$ and the target DNF $(x_1 \wedge x_2) \vee (x_3 \wedge x_4) \vee (x_5 \wedge x_6) \vee (x_7 \wedge x_8) \vee (x_9 \wedge x_10) \vee (x_{11} \wedge x_{12})$ we used $\eta = 10^{-3}$ for reducing the run time even more.

For validating if our reconstructed network is equivalent to $f^*$, we go over all the terms of $f^*$ and check that there is a neuron that is equal to it. In addition, we go over all the neurons and check that they have a corresponding term.

