# OpenReview forum: "On Learning Read-once DNFs With Neural Networks"
_ICLR.cc/2021/Conference — Reject_

### Official Review · AnonReviewer1 · 2020-10-25
**Review: On Learning Read-once DNFs With Neural Networks**

**Rating:** 5
**Confidence:** 4

**Review:**

This paper investigates the problem of learning monotone read-once DNF formulas using convex neural networks. Specifically, the authors explore the distribution-specific PAC setting, where training samples are drawn independently according to the uniform distributions and are labeled according to a target monotone read-once DNF. The main contribution of this study is essentially empirical: convex neural nets, trained with GD for minimizing the cumulative hinge loss, converge to global minima for which neural units coincide with the monomials of the target DNF. This remarkable stability is corroborated by theoretical insights about global minima.

First of all, the formal setting should be clarified: according to the specifications given in Section 3, this study focuses on “monotone” read-once DNF formula for which all literals are positive. I don’t think that this restriction has a major impact on the result since read-once DNFs are unate (i.e. we can rename negative literals in order to obtain a monotone variant).

Next, the learnability result about read-once DNF formula should be clarified. In the introduction, it is indicated that this concept class is efficiently PAC learnable under the uniform distribution, by quoting Fiat & Pechyony (2004). Well, this is not exactly true: Fiat & Pechyony (2004) used a result obtained by Mansour & Schain (2001), in which it was shown that read-once DNF formulas are properly and efficiently PAC learnable under any maximum entropy distribution. But in F&P (2004), it was not explicitly demonstrated that “any” read-once DNF formula is properly and efficiently PAC learnable under the uniform distribution. So, for the sake of completeness, it would be legitimate to provide such a result (in the Appendix, for example.)

Finally, and most importantly, I am not entirely convinced by the impact of this study. On the one hand, as indicated above, the authors empirically demonstrate that convex neural nets are able to learn monotone read-once DNF concepts by converging to global minima that capture the target concept. This is indeed interesting in practice, but there is no formal proof that convex neural nets are able to learn “any” monotone read-once DNF in polynomial time with polynomial sample complexity. On the other hand, it is already known that monotone read-once DNF functions are “improperly” but efficiently PAC learnable under the uniform distribution (Hancock & Mansour, 1991). In fact, Hancock and Mansour have shown that monotone read-k DNF functions are efficiently PAC learnable under product distributions. Actually, many subclasses of DNF are known to be efficiently learnable under product distributions, using spectral approaches (see e.g. Feldman, 2012). So, in light of such strong results, it seems that the contribution of the present paper is slightly behind.

Vitaly Feldman. Learning DNF expressions from Fourier spectrum. In Proc. Conf. Learn. Theory (COLT), pages 17.1–17.19, 2012.

Thomas Hancock and Yishay Mansour. Learning monotone k-µ DNF formulas on
product distributions. In Proceedings of the 4th Annual Conference on Computational
Learning Theory (COLT), pages 179–193, 1991.

Yishay Mansour and Mariano Schain, Learning with Maximum-Entropy Distributions. Machine
Learning, 45(2):123-145, 2001.

---

> ### Author Response · Authors · 2020-11-15
> **Novel results on the inductive bias of neural networks in challenging settings are significant regardless of stronger guarantees for other algorithms**
>
> Thank you for the thoughtful review. We address your comments below.
>
> ###
>
> “First of all, the formal setting should be clarified: according to the specifications given in Section 3, this study focuses on “monotone” read-once DNF formula for which all literals are positive. I don’t think that this restriction has a major impact on the result since read-once DNFs are unate (i.e. we can rename negative literals in order to obtain a monotone variant).”
>
> #####
>
>
> Thank you, we clarified this in the revised version. In the case of uniform distribution, you can assume unate DNFs WLOG. This follows since for the uniform distribution (that has IID Bernoulli(0.5) entries), by symmetry, any negated literal can be replaced with the original literal (without negation) and the results still hold (e.g., DNF recovery for the population case).
>
>
> ###
>
> “Next, the learnability result about read-once DNF formula should be clarified ... Fiat & Pechyony (2004) used a result obtained by Mansour & Schain (2001), in which it was shown that read-once DNF formulas are properly and efficiently PAC learnable under any maximum entropy distribution. But in F&P (2004), it was not explicitly demonstrated that “any” read-once DNF formula is properly and efficiently PAC learnable under the uniform distribution. ”
>
> #####
>
> Thank you, we fixed this in the revised version. We cited Mansour and Schain who showed learnability of any read-once DNF.
>
>
> ###
>
> “Finally, and most importantly, I am not entirely convinced by the impact of this study …. there is no formal proof that convex neural nets are able to learn “any” monotone read-once DNF in polynomial time with polynomial sample complexity. On the other hand, it is already known that monotone read-once DNF functions are “improperly” but efficiently PAC learnable under the uniform distribution (Hancock & Mansour, 1991). …. So, in light of such strong results, it seems that the contribution of the present paper is slightly behind.”
>
> #####
>
> Our main goal is to understand the inductive bias of GD on neural networks and *not* compare it to other algorithms. This is similar to other works on the inductive bias of neural networks which study networks on data that can be efficiently learned by other methods (e.g., Ji, Ziwei, and Matus Telgarsky. "Gradient descent aligns the layers of deep linear networks." ICLR. 2019 and Gunasekar, Suriya, et al. "Implicit bias of gradient descent on linear convolutional networks." Neurips 2018.‏). Furthermore, neural networks are usually very hard to analyze (compared to other algorithms) and therefore the difficulty of their analysis should be taken into account when assessing the significance of the results. Finally, due to the widespread use of deep-learning it is very important to understand in which cases such methods are provably correct.
>
> For example, consider the work of Ji and Telgarsky “Gradient descent aligns the layers of deep linear networks” published at ICLR 2019. They show that asymptotically gradient descent on a linear network converges to the maximum margin solution of linearly separable data. They do not show convergence rates. However, there exist other algorithms (e.g., SVM) that learn linearly separable data with convergence rates. Using an analogous argument to yours above, it could be claimed that the results of Ji and Telgarsky are not significant because there exists other algorithms (e.g., SVM) with stronger guarantees. However, we believe that this claim is incorrect. Indeed, Ji and Telgarsky study a *non-convex* problem which is much more difficult to analyze than SVM which is convex. Therefore, although they do not give convergence rates, their result is highly significant.
>
> In our case as well, there are other algorithms which can efficiently learn read-once DNFs. However, analyzing nonlinear networks in this setting is extremely challenging. Nonetheless, in this work, we make significant headway in this setting. Specifically, we gain a very good understanding of the inductive bias of GD via numerous theoretical and empirical results. We note that current works on neural networks, that go beyond the NTK analysis, consider either very simplified models (such as linear networks) or very simplified distributions such as linearly separable data. Here we consider a much more challenging setting of nonlinear models and nonlinear data. Therefore, we believe that our results are highly significant and can open up new research directions to understand neural networks in challenging settings.
>
> We hope that we have clarified the novelty and significance of our results. We kindly ask you to reevaluate your score based on our comments.

---

### Official Review · AnonReviewer4 · 2020-10-28
**Learning read-once DFNs using a convex neural network and gradient descent**

**Rating:** 7
**Confidence:** 2

**Review:**

In this paper the aim in to understand the inductive bias of neural networks learning DNFs. The focus is in convex neural networks and gradient descent. It is shown that under a symmetric initialization, the global minimum that gradient descent converges to is similar to a DNF-recovery solution. Further, experimental evaluation demonstrates that gradient descent can recover read-once DNFs from data.

Learning functions over Boolean variables is a fundamental problem and there have been an increasing interest towards using neural networks for this task. This paper sheds light to this task in a very specific case. I am not an expert on the area of the paper, and hence cannot fully assess the novelty and impact of the work. However, I found the development in the paper concise and enough details are provided for even a non-expert to follow the presentation. The restriction to read-once DNFs seems rather severe. However, I found the analysis of the inductive bias of gradient decent towards logical formulas interesting.


Pros:
1. Provides understanding of the inductive bias of convex neural networks using gradient descent
2. Computer assisted proof and experiments are used to complement theoretical results
3. Clearly and concisely written paper, a significant amount of supplementary material to  make a more detailed treatment available for those interested

Cons:
1. Results cover read-once DNFs which is very restricted class of DNFs, which might limit the impact of the results
2. Without access to code, it's impossible to assess the correctness and quality of the computer assisted proof


Questions:

It's quite unclear what is happening in Figure 2, could you add some more explanations?

Will the code related to computer assisted proof and the other experiments be made publicly available?

Minor comments:

Remark 3.1: "Boolean" not "boolean"

There are couple of references "We use a unique property of our setting that allows us to perform calculations in integers and avoid floating point errors." (intro, start of Sec. 6) where it is unclear what this property is. This is finally explained before Def. 6.3. It could be better to explain this in short in the earlier mentions, because in their current format they do not really help the reader, but rather make them wonder what is this property.

KKT in page 7 not defined

---

> ### Author Response · Authors · 2020-11-15
> **Thank you for the positive feedback. DNFs are hard in general, read-once DNFs setting is a good test-bed for understanding neural networks**
>
> Thank you for the positive feedback. We address your comments below.
>
> ###
>
> “Results cover read-once DNFs which is a very restricted class of DNFs, which might limit the impact of the results”
>
> #####
>
> As we discuss in the paper, DNFs in general are hard to learn efficiently in the general case. Thus, there is no reason to expect that they can be learned efficiently with neural nets. In contrast, read-once DNF is efficiently learnable (see citations in the main paper) and has been studied in many theoretical works. We therefore focus on this setting for understanding the important problem of learning Boolean functions with deep-learning. Finally, we note that already for read-once DNF, analyzing nonlinear networks is extremely challenging due to the non-convexity of the optimization problem and non-linearity of the ground-truth DNF. Nonetheless, we make significant progress in this setting and obtain novel theoretical and empirical results on the inductive bias of GD.
>
>
>
> ###
>
> “Without access to code, it's impossible to assess the correctness and quality of the computer assisted proof”
>
> #####
>
> We included the code in the submission. It is in the zip file in the supplementary material.
>
>
> ###
>
> “It's quite unclear what is happening in Figure 2, could you add some more explanations”
>
> #####
>
> Figure 2 shows the DNF recovery procedure that we describe in Section 6. It shows three sets of model parameters in the three sub-figures. In each sub-figure, each row in the image is a weight vector. Before plotting, we perform agglomerative clustering on the rows, so that similar weight vectors are plotted next to each other for clearer visibility of the clustering structure. The three figures are as follows. Figure 2(a) shows the global minimum that GD converges to. Figure 2b shows the same set of weights, but after the pruning step in Definition 6.1 (this mostly eliminates the “blueish” rows in 2a, but otherwise keeps the same three clear clusters in Figure 2a). Finally, we apply the beta-reconstruction procedure (Definition 6.2) to the weights in 2b to obtain the model in 2c. Recall that this procedure results in boolean weights, which is why Figure 2c contains only two colors.
>
> ###
>
> “KKT in page 7 not defined”
>
> #####
>
> KKT points are points which satisfy the Karush–Kuhn–Tucker optimality conditions which are *necessary* for any optimal solution. In Theorem 7.2, we show that the optimal solution (which is also a KKT point) is a DNF recovery solution.
>
> #################
>
> We fixed typos and clarified the text in the revised paper.

---

### Official Review · AnonReviewer2 · 2020-10-31
**The idea is interesting, but the results are preliminary.**

**Rating:** 4
**Confidence:** 3

**Review:**

The paper considers learning Boolean functions represented by read-once DNFs by using neural networks. The neural network architecture consists of a hidden layer with 2^D components, which is rich enough to express any Boolean functions. Given a whole 2^D instances of some read-once DNF, the authors showed that (1) weights corresponds to the true DNF is the global minimum of the loss minimization problem with the network, (2) they empirically observe that gradient descent with a rounding heuristics finds the true DNF expression, and(3) the solution of a 2-norm minimization recovers the true DNF.

The assumption that the whole set of instances of the true read-once DNF is given is too strong. It would be much nicer, given a partial set of instances S \subset X, one can learn a consistent DNF by using neural networks. Then, previous PAC learning results of read-once DNFs could be applied to obtain sample complexity results.

I don’t think the “computer-aided proof” is really a proof. So, I am afraid that it should not be stated as a theorem.

As a summary, I feel that the technical results are still preliminary and not mature enough to be published.

---

> ### Author Response · Authors · 2020-11-15
> **The population setting is common for understanding the inductive bias of neural networks**
>
> Thank you for the feedback. We address your concerns below.
>
> ###
>
> “The assumption that the whole set of instances of the true read-once DNF is given is too strong. It would be much nicer, given a partial set of instances S \subset X, one can learn a consistent DNF by using neural networks. Then, previous PAC learning results of read-once DNFs could be applied to obtain sample complexity results.”
>
> #####
>
> First, we note that it is common in deep-learning theory to analyze the population setting  (e.g, Daniely et al.,2020; Safran et al., 2018, citations are in the main paper). The rationale is that this often results in insight into behavior with smaller samples, and is more amenable to analysis. We also note that analyzing optimization in the small sample case is still a major open problem in deep-learning theory and there are very few cases where this is possible.
> Importantly, our population analysis reveals the inductive bias of the learning algorithm. Namely, that despite the ability to learn complex functions and the fact that the training set is large, GD in fact learns a simple model. This fact is robustly demonstrated in our experiments with smaller training samples, thus showing that the population analysis was useful for characterizing learning in the small-sample case as well. For example, in the experiment of Figure 3(c)  we successfully reconstructed the ground-truth DNF with 100 input variables from the network's weights - this is a novel and highly non-trivial achievement.
> We also derive several novel results for GD in the population setting (e.g., multiple global minima, computer assisted proof, preservation of terms in GD dynamics and characterization of min l2 solutions). All of these are challenging technically, and are an important step towards understanding learning of Boolean functions using deep-learning, which is an important open problem.
>
> ###
>
> “I don’t think the “computer-aided proof” is really a proof. So, I am afraid that it should not be stated as a theorem.”
>
> #####
>
> You claim that the computer assisted proof is not “really a proof” but do not provide any details. Computer assisted proofs are standard for proving theorems when the theoretical analysis is not tractable (e.g., as in Safran et al. (2018), the citation is in the main paper). Since our setting is deterministic and we perform calculations in integers, our computer assisted proof is exact and proves the stated claims in the theorem. We included the code for running the proof in the supplementary material.
>
>
>
>
> ################
>
> Finally, we would like to emphasize the difficulty of the problem we consider and the significance of our results. Current works on neural networks (that go beyond the NTK analysis) consider either very simplified models (such as linear networks) or very simplified distributions such as linearly separable data. Here we make significant headway in a much more challenging setting of nonlinear models and nonlinear data. Specifically, we gain a very good understanding of the inductive bias of GD in this case and verify our findings empirically in various experiments. Therefore, we believe that our results are significant and can open up new research directions to understand neural networks in challenging settings.
>
> We hope that we have clarified the novelty and significance of our results. We kindly ask you to reevaluate your score based on our comments.

---

### Decision · Program_Chairs · 2021-01-07
**Final Decision**

**Decision:**

Reject

**Comment:**

This paper studies how two-layer neural networks can learn DNFs. The paper provides some theoretical analysis together with empirical evidence.

The direction of analyzing how neural networks learn certain concept classes is definitely extremely important, and the authors do make some progress towards this direction. However, there are some major concerns about the paper:

 In the main result, the authors seem to only able to prove that the learning process converges with exponentially many neurons (exponential in the input dimension, see 6.1 setup). With this many neurons, it is unclear whether the result is directly covered by existing works such as:



(a). "Learning and generalization in overparameterized neural networks, going beyond two layers".
(b). "Fine-grained analysis of optimization and generalization for overparameterized two-layer neural networks"

These two works provide efficient optimization and generalization bounds w.r.t the complexity of the function. However, these works are still in the NTK regime. It would be nice if the authors can distinguish the current technique from NTKs by providing some theoretical guarantee that their main result is indeed more efficient than kernels (as they argue in the intro), the author can refer to:



(a) "What can ResNet efficiently learn, going beyond kernels" .
 (b) "When Do Neural Networks Outperform Kernel Methods?"


With the current form of the draft, it is unclear how the result is better than existing approaches. The authors should address that in the next version of the paper.

Missing reference of NTKs:
"A convergence theory for deep learning via over-parameterization"